# Iterative Preference Optimization with Proximal Policy Regularization for Large Language Model Alignment

**Siyuan Xu**                                                                    *spx5032@psu.edu*
*The Pennsylvania State University*

**Hangfan Zhang**                                                                *hbz5148@psu.edu*
*The Pennsylvania State University*

**Zhimeng Guo**                                                                  *zhimeng@psu.edu*
*The Pennsylvania State University*

**Huaisheng Zhu**                                                                *hvz5312@psu.edu*
*The Pennsylvania State University*

**Yue Mao**                                                                      *ypm5140@psu.edu*
*The Pennsylvania State University*

**Shicheng Liu**[*]                                                              *sfl5339@psu.edu*
*The Pennsylvania State University*

**Reviewed on OpenReview:** *https://openreview.net/forum?id=xox05Tr4Vh*

## Abstract

Aligning large language models (LLMs) with human preferences is commonly achieved via supervised fine-tuning followed by preference optimization. While direct preference optimization (DPO) offers a simple and efficient alternative to RLHF, its offline and off-policy nature can induce a distribution shift between the policy used to sample preference pairs and the continually updated policy being optimized, reducing data efficiency and limiting alignment gains. We propose *Iterative Proximal Policy Regularized Preference Optimization* (Iterative PRPO), which introduces a proximal regularization that explicitly constrains the optimized policy to remain close to the sampling policy within each iteration, thereby mitigating distribution shift while preserving the efficiency of DPO-style updates. Starting from an RLHF objective with a KL constraint to the sampling policy, we derive an equivalent direct preference optimization formulation that requires offline preference pairs under the sampling policy. Across summarization and dialogue alignment benchmarks, Iterative PRPO consistently improves win rates over offline DPO and iterative DPO baselines under both reward-model and GPT-4o evaluations, with comparable computational cost. Moreover, the same proximal regularization principle generalizes to advanced preference optimization objectives, including Identity Preference Optimization (IPO), self-play preference optimization (SPPO), and efficient exact optimization (EXO), yielding Iterative PR-IPO, PR-SPPO, and PR-EXO variants that further strengthen alignment across model scales.

## 1 Introduction

Substantial progress in pre-trained large language models (LLMs) has underscored the growing importance of a critical post-training phase known as alignment with human preferences (Ouyang et al., 2022; Rafailov et al., 2023; Xiao et al., 2025). The current alignment approach can be unfolded into two stages: (i) supervised

---

[*]Corresponding author.

Table 1: Motivations for the proposed algorithm, which mitigates distribution shift while maintaining high efficiency.

| Method | Distribution Shift Issue |
|---|---|
| RLHF | Not present (on-policy sampling) |
| (Iterative) DPO variants | Present (within-iteration; partially mitigated across iterations by data regeneration) |
| Iterative PRPO (ours) | Mitigated (off-policy sampling under the policy optimization with the proximal policy regularization) |

fine-tuning stage, which uses demonstration data containing demonstrated completions to prompts, and (ii) preference optimization, which exploits human preference data containing preferred and rejected completions to prompts. The alignment phase substantially enhances the performance of LLMs, particularly in complex instruction-following tasks such as knowledge-based question answering (Bai et al., 2022), summarization (Stiennon et al., 2020), and mathematical problem solving (Cobbe et al., 2021).

The mainstream approaches for preference optimization can be categorized into (i) reinforcement learning from human feedback (RLHF) (Christiano et al., 2017; Ouyang et al., 2022) and (ii) direct preference optimization (DPO) (Rafailov et al., 2023; Ji et al., 2024; Liu et al., 2024). RLHF approaches first train a reward model from human-labeled preference data, then perform reinforcement learning (RL) with the reward signals produced by the learned reward model. The RL phase typically involves on-policy sampling, i.e., a batch of prompt-completion pairs is collected through model inference and is used for the model parameter update at each training step. In contrast, DPO approaches are offline, which aim to optimize preference probabilities on a fixed preference dataset. DPO approaches have gained popularity due to **their inherent simplicity and efficiency** compared to RLHF. However, recent results have demonstrated that **on-policy sampling in RLHF plays a crucial role for policy improvement** (Tajwar et al., 2024; DeepSeek-AI, 2025), particularly when the distribution shift from the reference model to the optimal model is large (Tajwar et al., 2024).

To make preference data sampling more aligned with the on-policy setting and mitigate the distribution shift between the sampling policy and the optimized policy observed in offline preference optimization, iterative DPO variants have been proposed (Calandriello et al., 2024; Dong et al., 2023; Xiong et al., 2024; Wu et al., 2024; Pang et al., 2024). Specifically, these methods periodically update the model parameters and regenerate the preference dataset by model inference from the updated policy. However, although the iterative variants of DPO refresh the preference data in each iteration, due to the inherent nature of the off-policy sampling, the distribution shift issue still exists. Specifically, within each iteration, after multiple optimization steps, the current policy may deviate substantially from the policy used to generate the preference data at the beginning of the iteration. Moreover, the number of optimization steps in each iteration is required to be manually determined. Existing work typically adopts a large number, usually completing the optimization of more than ten thousand pairs of preference data in merely three iterations (Pang et al., 2024; Wu et al., 2024), which exacerbates the distribution shift issue.

To address this issue, we propose an *iterative proximal policy regularized preference optimization* (iterative PRPO) algorithm, which incorporates trust-region/proximal policy regularization motivated by RL algorithms (Schulman et al., 2015; 2017) into iterative preference optimization. Specifically, in each preference optimization iteration, a proximal policy regularization is imposed on the standard RLHF objective to prevent the policy from being optimized far from the sampling policy. Then we derive a direct preference optimization algorithm that is mathematically equivalent to the optimization problem of RLHF with the proximal policy regularization. Note that the proposed iterative PRPO method follows the same procedures of the data sampling and the policy optimization as existing iterative DPO methods, thereby retaining their high computational efficiency.

**Main contribution.** In this paper, we propose the iterative proximal policy regularized preference optimization (iterative PRPO) algorithm, an efficient iterative preference optimization method that mitigates

distribution shift while preserving the high computational efficiency of existing iterative preference optimization approaches. Specifically, the iterative PRPO incorporates a proximal policy regularization into the standard RLHF objective and derives an equivalent direct preference optimization formulation, enabling preference optimization that remains close to the sampling policy within each iteration. This effectively reduces the distribution shift caused by off-policy sampling, especially when a large number of optimization steps are applied per iteration, leading to improved alignment performance. Furthermore, the iterative PRPO framework is general and can be seamlessly integrated into a wide range of advanced preference optimization methods, resulting in consistent performance improvements across diverse tasks.

**Experiments.** We conduct extensive experiments with the proposed iterative PRPO algorithm across a variety of preference optimization tasks and observe consistent improvements over existing iterative preference optimization methods, while maintaining comparable computational costs. Our results demonstrate that incorporating proximal policy regularization effectively reduces distribution shift within each optimization iteration while simultaneously enhancing alignment performance. Finally, we show that applying the proposed proximal regularization framework to existing iterative DPO variants generally yields superior results compared to their counterparts without regularization.

## 2 Related Works

**Offline DPO** eliminates the need for explicit reward modeling in RLHF by directly optimizing the policy using preference data, effectively integrating implicit reward estimation into the policy optimization process. The core insight of Direct Preference Optimization (DPO) (Rafailov et al., 2023) is that, under a Bradley–Terry style preference model (Bradley & Terry, 1952), the optimal policy for the KL-regularized RLHF objective admits a closed-form solution that can be directly optimized via supervised learning. This formulation bypasses explicit reward model training and on-policy rollouts, significantly simplifying the alignment pipeline while retaining strong empirical performance. Building on this paradigm, a growing body of DPO-like methods has been proposed to improve robustness, stability, and theoretical grounding. Identity Preference Optimization (IPO) (Azar et al., 2024) reframes preference learning through a more general theoretical lens, clarifying the connection between preference optimization and imitation learning. Efficient exact optimization (EXO) (Ji et al., 2024) studies exact optimization objectives for alignment and provides efficient algorithms with improved convergence guarantees. KTO (Ethayarajh et al., 2024) introduces a prospect-theoretic formulation that accounts for asymmetric human risk preferences, while RRHF (Yuan et al., 2023), Cal-DPO (Xiao et al., 2024), and SimPO (Meng et al., 2024) explore alternative ranking losses, calibration strategies, and reference-free objectives to mitigate issues such as reward hacking and sensitivity to reference models. Despite their effectiveness, these offline methods fundamentally rely on fixed, off-policy preference datasets, making them vulnerable to distribution mismatch when iteratively updating the policy.

**Iterative preference optimization** methods address this limitation by alternating between policy updates and preference data regeneration using the current model. This paradigm more closely resembles online RLHF (Christiano et al., 2017; Ouyang et al., 2022) while retaining the computational efficiency of preference-based optimization. Several variants have been developed, either by extending existing offline methods or by proposing new iterative formulations (Calandriello et al., 2024; Dong et al., 2023; Xiong et al., 2024; Wu et al., 2024; Pang et al., 2024). For example, building on offline DPO (Rafailov et al., 2023), Pang et al. (2024) proposes Iterative DPO to improve mathematical reasoning, showing that repeatedly refreshing preference data can substantially enhance reasoning depth. Calandriello et al. (2024) extends IPO (Azar et al., 2024) to an online setting, introducing regularization strategies to stabilize preference updates and prevent overfitting to newly collected data. RAFT (Dong et al., 2023) combines reward-ranked fine-tuning with iterative preference collection, while Xiong et al. (2024) provide a principled analysis of iterative RLHF under KL constraints, bridging theoretical guarantees and practical algorithms. More recently, self-play preference optimization (SPPO) (Wu et al., 2024) formulates alignment as a zero-sum game between policies, deriving optimal strategies via Nash equilibria. While these methods reduce distribution mismatch, they often still lack explicit mechanisms to control policy drift between iterations. Among these, the analysis of Xiong et al. (2024) is closest to our work: they study iterative RLHF under a single KL constraint to the reference model $\pi_{\text{ref}}$ and derive regret/sample-complexity bounds for the on-policy case, but their algorithmic prescription still reduces to DPO-style updates with the standard $\pi_{\text{ref}}$ as the sole reference. In

contrast, our PRPO objective introduces a *second* KL constraint to the within-iteration sampling policy $\pi_{\text{sample}}$ (Eq. 5), and applying Lagrangian duality to this two-constraint problem yields the closed-form optimum with a *geometric-mixture reference* $\pi_{\text{ref}}^{\alpha} \pi_{\text{sample}}^{1-\alpha}$ (Theorem 1), a structure that does not appear in Xiong et al. (2024)'s derivation. We also note that Cal-DPO (Xiao et al., 2024) addresses sensitivity to $\pi_{\text{ref}}$ via reward calibration, while RAFT (Dong et al., 2023) re-samples on-policy completions but still optimizes the standard DPO loss; in both cases, the reference distribution itself is not modified to track the sampling policy, which is the distinguishing feature of PRPO.

**Proximal policy regularization**, as introduced in trust region policy optimization (TRPO) (Schulman et al., 2015), is designed to limit large distribution shifts between the data-collecting policy and the updated policy by enforcing a hard KL constraint. This principle is crucial for stabilizing policy gradient methods, as large policy updates can invalidate gradient estimates. Proximal policy optimization (PPO) (Schulman et al., 2017; Ouyang et al., 2022) provides a first-order approximation to TRPO and has become the de facto algorithm for RLHF due to its empirical stability and scalability. Although DPO and its iterative variants are derived as approximations to the optimal KL-regularized RLHF solution (Rafailov et al., 2023), they fundamentally operate in an off-policy regime. As a result, they remain susceptible to distribution shift when the learned policy deviates significantly from the data-generating policy, especially in iterative settings (Tajwar et al., 2024). In contrast, trust-region methods explicitly constrain policy updates to remain within a local neighborhood, offering strong stability guarantees. Motivated by these observations, we propose a new iterative preference optimization algorithm, *iterative PRPO*, which integrates the distribution shift-preventing principles of TRPO into the preference optimization framework. Our approach preserves the computational efficiency and simplicity of DPO-style objectives while introducing proximal regularization to stabilize iterative updates, bridging the gap between offline preference optimization and trust-region-based RLHF. Concretely, PPO implements trust-region behavior within a policy optimization step via importance-ratio clipping, optimizing a surrogate of the form $\mathbb{E}[\min(r_t(\theta)\,\hat{A}_t, \text{clip}(r_t(\theta), 1 - \varepsilon, 1 + \varepsilon)\,\hat{A}_t)]$ with $r_t(\theta) = \pi_\theta(a_t \mid s_t)/\pi_{\theta_{\text{old}}}(a_t \mid s_t)$, which requires on-policy rollouts and a learned value function. PRPO instead bakes the trust-region idea into the *closed-form solution* of the constrained RLHF objective, yielding a DPO-style supervised loss with a modified reference distribution $\pi_{\text{ref}}^{\alpha} \pi_{\text{sample}}^{1-\alpha}$. PRPO therefore inherits PPO's stability motivation but retains DPO's offline, gradient-stable, RL-free training pipeline.

## 3 Preliminaries

Let the token sequence $x = [x_1, x_2, \ldots]$ represent the input prompt, and $y = [y_1, y_2, \ldots]$ represent the generated completion. Denote the completion generation policy of the LLM as $\pi_\theta$, where $\theta$ are the model parameters and $\pi_\theta(y|x)$ denotes the probability of generating completion $y$ conditional on the input $x$. Specifically, given an input $x$, the LLM generates the completion autoregressively, where at each time step $t$, a token is sampled as $y_t \sim \pi_\theta(\cdot|y_1, \ldots, y_{t-1}, x)$. This process continues until an end-of-sentence (EOS) token is generated or a predefined maximum length $T$ is reached.

**Preference optimization problem.** Consider the problem of preference optimization. Let $y_w$ and $y_l$ denote two completions, sampled from a reference policy $\pi_{\text{ref}}(y|x)$. The completion pairs are then queried to a preference oracle to obtain the preference label, denoted as $y_w \succ y_l | x$, where $y_w$ and $y_l$ denote chosen and rejected responses, respectively. Collect the preference pairs $(x, y_w, y_l)$ as the dataset $\mathcal{D}$. The goal is to learn a generation policy $\pi_\theta$ for aligning human preferences.

**RLHF.** RLHF can be unfolded into two stages: (i) the reward modeling stage and (ii) the RL stage. In the stage of reward modeling, a reward model $r_\phi$ is learned from the preference dataset $\mathcal{D}$ as a surrogate to expensive human labeling. Specifically, the reward model takes a prompt-completion pair as the input and produces a reward value to reflect the preference likelihood for completion given the prompt. The preference likelihood is modeled by the Bradley-Terry model (Bradley & Terry, 1952), i.e., $\mathbb{P}(y_w \succ y_l | x) = \sigma(r(x, y_w) - r(x, y_l))$, and the reward model is trained by maximizing the log-likelihood, i.e., minimizing the loss function $\mathcal{L}_r(r_\phi; \mathcal{D})$:

$$\mathbb{E}_{(x, y_w, y_l) \sim \mathcal{D}} \left[ \log \sigma \left( r_\phi(x, y_w) - r_\phi(x, y_l) \right) \right], \tag{1}$$

where $\sigma$ is the logistic function. In the RL stage, the policy $\pi_\theta$ aims to solve the following problem:

$$\max_{\pi_\theta} \mathbb{E}_{x \sim \mathcal{D}, y \sim \pi_\theta(\cdot|x)}[r_\phi(x, y)] - \beta \mathbb{E}_{x \sim \mathcal{D}}[\mathbb{D}_{\mathrm{KL}}(\pi_\theta(\cdot|x)||\pi_{\mathrm{ref}}(\cdot|x))], \tag{2}$$

where $\beta$ is a hyper-parameter controlling the deviation of the learned policy $\pi_\theta$ from the reference model $\pi_{\mathrm{ref}}$, and $\mathbb{D}_{\mathrm{KL}}(\pi_\theta(\cdot|x)||\pi_{\mathrm{ref}}(\cdot|x)) \triangleq E_{y \sim \pi_\theta(\cdot|x)}[\log \pi_\theta(y|x) - \log \pi_{\mathrm{ref}}(y|x)]$ is the KL divergence between $\pi_\theta$ and $\pi_{\mathrm{ref}}$.

**DPO.** DPO eliminates the need for the reward modeling stage in RLHF. Rafailov et al. (2023) is one of the most popular DPO approaches, which first uses the log-likelihood of the policy to represent the reward function via a closed-form solution of the RLHF optimization problem in equation 2:

$$r(x, y) = \beta \log \frac{\pi_\theta(y|x)}{\pi_{\mathrm{ref}}(y|x)} + \beta \log Z(x), \tag{3}$$

where $Z(x)$ is the partition function. Similar to equation 1, the surrogate log-likelihood under the BT model is maximized, i.e., minimizing the loss function:

$$\mathcal{L}_{\mathrm{DPO}}(\pi_\theta; \mathcal{D}) = -\mathbb{E}_{(x, y_w, y_l) \sim \mathcal{D}}\left[\log \sigma\left(\beta \log \frac{\pi_\theta(y_w|x)}{\pi_{\mathrm{ref}}(y_w|x)} - \beta \log \frac{\pi_\theta(y_l|x)}{\pi_{\mathrm{ref}}(y_l|x)}\right)\right], \tag{4}$$

which cancels the partition function $Z(x)$ in equation 3 during the preference optimization.

**Iterative preference optimization.** Iterative preference optimization repeats optimizing the model parameters using the offline preference optimization method, such as DPO, and generating preference data by performing model inference using the updated policy. Specifically, at the $n$-th iteration, the following steps are implemented:

(i) Perform model inference using the policy $\pi_{\theta_{n-1}}$ produced from the $n$-th iteration to generate at least two completions, $y^{(1)}$ and $y^{(2)}$, for each prompt $x$ in the dataset $\mathcal{D}$.

(ii) Label the preference between each pair $(x, y^{(1)}, y^{(2)})$ to obtain a preferred and less-preferred completion, denoted as $(x, y_w, y_l)$, which forms the labeled dataset $\mathcal{D}^{(n)}$.

(iii) Optimize the policy on $\mathcal{D}^{(n)}$ using the DPO loss function in equation 4 or its variants. The optimization starts from the policy $\pi_{\theta_{n-1}}$ and yields an updated policy $\pi_{\theta_n}$.

Note that in step (ii), the preference label can be provided either by human annotators or by a reward model trained in a manner similar to RLHF.

# 4 Proposed Method

We introduce the iterative proximal policy regularized preference optimization (iterative PRPO) algorithm, an efficient iterative preference optimization method that mitigates distribution shift while preserving the high computational efficiency of existing iterative DPO approaches. We introduce the derivation of the PRPO method in Section 4.1 and show the analysis and discussion in Section 4.2. In Section 4.3, we integrate the PRPO into the iterative preference optimization. In Section 4.4, we extend the proximal policy regularization framework to several advanced preference optimization methods, including IPO (Azar et al., 2024), SPPO (Wu et al., 2024), and EXO (Ji et al., 2024).

## 4.1 Proximal Policy Regularized Preference Optimization

Before initiating preference optimization (Rafailov et al., 2023), the preference data $\mathcal{D}_{\mathrm{sample}}$ is generated by a policy $\pi_{\mathrm{sample}}$. The distribution shift issue will appear when implementing the preference optimization using $\mathcal{D}_{\mathrm{sample}}$. Specifically, when the data distribution produced by the optimized policy $\pi_{\mathrm{opt}}$ is significantly deviated from that produced by $\pi_{\mathrm{sample}}$, such that the preference data $\mathcal{D}_{\mathrm{sample}}$ may no longer provide effective

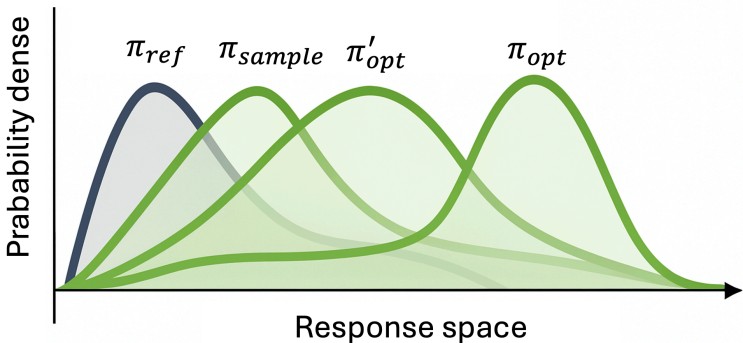

Figure 1: Illustration of the distribution shift issue in DPO approaches. Consider the policy $\pi$ is expected to be optimized to the policy $\pi_{\mathrm{opt}}$. Since the preference data $\mathcal{D}_{\mathrm{sample}}$ is collected by $\pi_1$, when the policy is shifted to $\pi'_{\mathrm{opt}}$, $\mathcal{D}_{\mathrm{sample}}$ cannot serve as good supervision for $\pi'_{\mathrm{opt}}$ to go to $\pi_{\mathrm{opt}}$. Consequently, the probability density peak of $\pi'_{\mathrm{opt}}$ will continue to increase in regions unsupported by the original data, causing the policy to deviate further from the desired target $\pi_{\mathrm{opt}}$.

supervision for optimizing $\pi_{\mathrm{opt}}$. This misalignment can lead to unintended consequences, such as the model assigning higher probabilities to responses that were not favored in the original preference data, thereby deviating from the intended optimization policy. Figure 1 illustrates the effects of such distribution shifts.

To address the issue, we start from the original objective of DPO, the objective of RLHF in equation 2, with the proximal policy regularization constraint:

$$
\max_{\pi_\theta} \ \mathbb{E}_{x\sim\mathcal{D},y\sim\pi_\theta(\cdot|x)}[r_\phi(x,y)] - \beta_0 \ \mathbb{E}_{x\sim\mathcal{D}}[\mathbb{D}_{\mathrm{KL}}(\pi_\theta(\cdot|x)\|\pi_{\mathrm{ref}}(\cdot|x))]
$$
$$
\text{s.t.} \ \mathbb{E}_{x\sim\mathcal{D}}[\mathbb{D}_{\mathrm{KL}}(\pi_\theta(\cdot|x)\|\pi_{\mathrm{sample}}(\cdot|x))] \leq \epsilon.
\tag{5}
$$

Based on equation 2, the optimization problem equation 5 imposes $\mathbb{E}_{x\sim\mathcal{D}}[\mathbb{D}_{\mathrm{KL}}(\pi_\theta(\cdot|x)\|\pi_{\mathrm{sample}}(\cdot|x))] \leq \epsilon$, which constrains the deviation of the optimized policy $\pi_\theta$ from the sampling policy $\pi_{\mathrm{sample}}$ to avoid the distribution shift.

We aim to solve the optimization problem equation 5 in the manner of direct preference optimization, i.e., solving using the offline preference data sampled by $\pi_{\mathrm{sample}}$, which is denoted as $\mathcal{D}_{\mathrm{sample}}$. We first derive the closed-form solution of equation 5.

**Theorem 1.** *There exists a constant $\beta \geq \beta_0$ such that the optimal solution $\pi^*_{r_\phi}$ of problem equation 5 takes the form of $\pi^*_{r_\phi}(y|x) =$*

$$
\frac{1}{Z(x)}\pi^\alpha_{ref}(y|x)\pi^{1-\alpha}_{sample}(y|x)\exp(\frac{1}{\beta}r_\phi(x,y)),
$$

*where $\alpha = \beta_0/\beta \leq 1$ and $Z(x) = \sum_y \pi^\alpha_{ref}(y|x) \ \pi^{1-\alpha}_{sample}(y|x)\exp(\frac{1}{\beta}r_\phi(x,y))$ is the partition function.*

See Appendix A.1.1 for the proof. With the closed-form solution in Theorem 1, we can express the reward function $r_\phi$ in terms of its corresponding optimal policy $\pi^*_{r_\phi}$. Specifically, we have $r_\phi(x,y) =$

$$
\beta \log \frac{\pi^*_{r_\phi}(y|x)}{\pi^\alpha_{\mathrm{ref}}(y|x)\pi^{1-\alpha}_{\mathrm{sample}}(y|x)} + \beta \log Z(x).
\tag{6}
$$

We apply the parameterization of policy $\pi^*_{r_\phi}$ in equation 6 to the Bradley-Terry preference model, we can obtain the likelihood of the preference

$$
\mathbb{P}\left(y_w \succ y_l|x\right) = \sigma\left(\beta \log \frac{\pi^*_{r_\phi}(y_w|x)}{\pi^\alpha_{\mathrm{ref}}(y_w|x)\pi^{1-\alpha}_{\mathrm{sample}}(y_w|x)} - \beta \log \frac{\pi^*_{r_\phi}(y_l|x)}{\pi^\alpha_{\mathrm{ref}}(y_l|x)\pi^{1-\alpha}_{\mathrm{sample}}(y_l|x)}\right).
$$

Therefore, the PRPO loss function is to minimize the negative log-likelihood of the preferences, i.e., $\mathcal{L}_{\mathrm{PRPO}}\left(\pi_\theta; \mathcal{D}_{\mathrm{sample}}\right) =$

$$-\mathbb{E}_{(x,y_w,y_l)\sim\mathcal{D}_{\mathrm{sample}}}\left[\log\sigma\left(\beta\log\frac{\pi_\theta(y_w|x)}{\pi_{\mathrm{ref}}^\alpha(y_w|x)\pi_{\mathrm{sample}}^{1-\alpha}(y_w|x)} - \beta\log\frac{\pi_\theta(y_l|x)}{\pi_{\mathrm{ref}}^\alpha(y_l|x)\pi_{\mathrm{sample}}^{1-\alpha}(y_l|x)}\right)\right], \qquad (7)$$

where the preference dataset $\mathcal{D}_{\mathrm{sample}}$ is sampled from the policy $\pi_{\mathrm{sample}}$ and then labeled.

## 4.2 Analysis and Discussion

In this section, we provide the analysis of the proposed PRPO method. We denote that

$$f_\theta(\pi) = \log\frac{\pi_\theta(y_w|x)}{\pi(y_w|x)} - \log\frac{\pi_\theta(y_l|x)}{\pi(y_l|x)}.$$

The PRPO loss function can be rewritten as

$$\mathcal{L}_{\mathrm{PRPO}}\left(\pi_\theta; \mathcal{D}_{\mathrm{sample}}\right) = -\mathbb{E}_{(x,y_w,y_l)\sim\mathcal{D}_{\mathrm{sample}}}\left[\log\sigma\left(\beta\alpha f_\theta(\pi_{\mathrm{ref}}) + \beta(1-\alpha)f_\theta(\pi_{\mathrm{sample}})\right)\right],$$

and the DPO loss function can be written as

$$\mathcal{L}_{\mathrm{DPO}}\left(\pi_\theta; \mathcal{D}_{\mathrm{sample}}\right) = -\mathbb{E}_{(x,y_w,y_l)\sim\mathcal{D}_{\mathrm{sample}}}\left[\log\sigma\left(\beta f_\theta(\pi_{\mathrm{ref}})\right)\right].$$

Note that the sigmoid function $\log\sigma(\cdot)$ is monotonically increasing, and therefore both the DPO and PRPO loss functions encourage higher probability for the preferred completion $\pi_\theta(y_w|x)$ and lower probability for the less preferred one $\pi_\theta(y_l|x)$. On the other hand, the gradient norm of $\log\sigma(\cdot)$ is monotonically decreasing, which naturally prevents the argument from becoming excessively large. As a result, the PRPO loss attempts to penalize the deviation of $\pi_\theta$ from $\pi_{\mathrm{ref}}$ and $\pi_{\mathrm{sample}}$ simultaneously. In particular, if the combined PRPO margin $\alpha f_\theta(\pi_{\mathrm{ref}}) + (1-\alpha)f_\theta(\pi_{\mathrm{sample}})$ becomes large, the gradient contribution from the logistic loss tends toward zero, softly attenuating the update. Here, the reference policy $\pi_{\mathrm{ref}}$ incorporates the capabilities obtained during LLM pretraining, while $\pi_{\mathrm{sample}}$ plays a critical role in mitigating distribution shift during preference optimization. This large-margin gradient attenuation is formalized in Proposition 1 (Appendix A.1.2).

**Relationship to PPO.** Here we show the relationship between the proposed PRPO and Proximal Policy Optimization (PPO) (Schulman et al., 2017). The two methods share the same high-level motivation, i.e., preventing the policy from deviating too far during an update step by the trust-region constraint, i.e., but differ fundamentally in how this is achieved. PPO implements the trust-region constraint *within an RL loop* via importance-ratio clipping, optimizing a surrogate objective $\mathbb{E}[\min(r_t(\theta)\hat{A}_t, \mathrm{clip}(r_t(\theta), 1-\varepsilon, 1+\varepsilon)\hat{A}_t)]$, which requires on-policy rollouts and a learned value function. PRPO, by contrast, encodes the proximity constraint to $\pi_{\mathrm{sample}}$ directly into the *closed-form solution* of the constrained RLHF objective, yielding a DPO-style supervised loss (Eq. equation 7) with a modified reference distribution $\pi_{\mathrm{ref}}^\alpha\pi_{\mathrm{sample}}^{1-\alpha}$. PRPO therefore inherits PPO's stability motivation while retaining DPO's offline, RL-free training pipeline.

Note that the PRPO loss equation 7 is reduced to the DPO loss in equation 4, when the sampling policy $\pi_{\mathrm{sample}}$ is the reference model $\pi_{\mathrm{ref}}$. When the sampling policy is significantly deviated from the reference model $\pi_{\mathrm{ref}}$, the training dataset $\mathcal{D}_{\mathrm{sample}}$ along with the sampling policy $\pi_{\mathrm{sample}}$ is ignored, which leads to a mismatch issue. Specifically, although the DPO equation 4 aims to recover the optimal policy of RLHF, it requires an implicit assumption that the preference dataset can cover a sufficient response space and can include the responses generated by the optimal policy. Given that the training dataset $\mathcal{D}_{\mathrm{sample}}$ is only sampled by $\pi_{\mathrm{sample}}$, the PRPO loss in equation 7 explicitly accounts for the deviation of the optimized policy from the sampling policy $\pi_{\mathrm{sample}}$ to mitigate the mismatch issue of DPO.

## 4.3 Integrate PRPO to Iterative Preference Optimization

In this section, we integrate PRPO into the iterative preference optimization framework and demonstrate its advantages in this setting.

---

**Algorithm 1** Iterative PRPO algorithm

---

**Require:** Initial policy $\pi_{\theta_0} = \pi_{\mathrm{ref}}$; the dataset of prompts $\mathcal{D}$; a preference dataset $\mathcal{D}_{\mathrm{pref}}$.

 1: Train a reward model $r_\phi$ using $\mathcal{D}_{\mathrm{pref}}$
 2: **for** $n = 1$ to $N$ **do**
 3:    **(Inference)** For each prompt $x \in \mathcal{D}$, use policy $\pi_{\theta_{n-1}}$ to generate completions: $y^{(1)}$, $y^{(2)}$
 4:    **(Labeling)** For each $(x, y^{(1)}, y^{(2)})$, obtain a preference label to identify preferred $y_w$ and less-preferred $y_l$ by the reward model $r_\phi$, forming dataset $\mathcal{D}^{(n)} = \{(x, y_w, y_l)\}$
 5:    **(Optimization)** Minimize the PRPO loss in equation 7 on $\mathcal{D}^{(n)}$, i.e., $\mathcal{L}_{\mathrm{PRPO}}(\pi_\theta; \mathcal{D}^{(n)})$, starting from $\pi_{\theta_{n-1}}$, resulting in updated policy $\pi_{\theta_n}$
 6: **end for**
 7: **return** $\pi_{\theta_N}$

---

At the $n$-iteration of the iterative preference optimization, the preference data $\mathcal{D}^{(n)}$ is sampled by the policy $\pi_{\theta_{n-1}}$, i.e., the policy optimized by the last iteration. The sampling policy is usually significantly deviated from $\pi_{\theta_0}$ when $n$ is large. As analyzed in Section 3, the PRPO is better equipped to handle this scenario by explicitly accounting for the evolving sampling distribution during the iterative preference optimization.

**Algorithm statement.** We summarize the iterative PRPO algorithm in Algorithm 1. In practice, the preference labeling in line 4 can be done by human annotators or by a well-trained reward model. In this paper, we consider an experimental setting that is fully consistent with offline preference optimization, such as (offline) DPO. Specifically, only the preference dataset $\mathcal{D}_{\mathrm{pref}}$ is given, and external reward models and human annotators are not accessible. Therefore, in line 1, we first train a reward model $r_\phi$ using $\mathcal{D}_{\mathrm{pref}}$ for the preference labeling in line 4 during the optimization iterations.

**Computational cost analysis.** The procedure of the iterative PRPO algorithm closely resembles that of the iterative DPO, and the computational cost is similarly comparable. The primary difference lies in the additional requirement for PRPO to compute $\pi_{\theta_{n-1}}(y_w|x)$ and $\pi_{\theta_{n-1}}(y_l|x)$. Regarding GPU memory usage, the extra model $\pi_{\theta_{n-1}}$ must be loaded onto the GPU. However, since this model is used solely for probability evaluation, without gradient computation, its memory cost is significantly smaller than that of the optimized policy $\pi_\theta$, often requiring only about one-tenth as much memory. As a result, the overall GPU memory cost remains comparable. In terms of time complexity, the evaluation of $\pi_{\theta_{n-1}}(y_w|x)$ and $\pi_{\theta_{n-1}}(y_l|x)$ does not involve auto-regressive decoding or gradient backpropagation, making it substantially faster than both model inference and gradient-based optimization. As a result, the overall time complexity remains similar.

### 4.4 Extension to IPO, SPPO, and EXO

Several advanced preference optimization algorithms, including IPO (Azar et al., 2024), SPPO (Wu et al., 2024), and EXO (Ji et al., 2024), are proposed to improve DPO. However, none of them explicitly address the data distribution shift issue of DPO. To handle the problem, we extend the iterative PRPO framework to these methods.

We first compare the loss functions in the above algorithms. Denote that

$$a = \beta \log \frac{\pi_\theta(y_w|x)}{\pi_{\mathrm{ref}}(y_w|x)}, \quad b = \beta \log \frac{\pi_\theta(y_l|x)}{\pi_{\mathrm{ref}}(y_l|x)}.$$

Then, the losses on a single preference data pair $(y_w, y_l, x)$ can be written as

$$\ell_{\mathrm{DPO}}(y_w, y_l, x) = -\log \sigma(a - b),$$
$$\ell_{\mathrm{IPO}}(y_w, y_l, x) = [(a - b) - 1]^2,$$
$$\ell_{\mathrm{SPPO}}(y_w, y_l, x) = (a - 1/2)^2 + (b + 1/2)^2,$$
$$\ell_{\mathrm{EXO}}(y_w, y_l, x) = \sigma(a - b) \log \frac{\sigma(a - b)}{1 - \epsilon} + \sigma(b - a) \log \frac{\sigma(b - a)}{\epsilon}.$$

The proposed PRPO loss can be written as

$$\ell_{\mathrm{PRPO}}\left(y_w, y_l, x\right) = -\log \sigma(a' - b')$$

where

$$a' = \beta \log \frac{\pi_\theta(y_w|x)}{\pi_{\mathrm{ref}}^\alpha(y_w|x)\pi_{\mathrm{sample}}^{1-\alpha}(y_w|x)}, \quad b' = \beta \log \frac{\pi_\theta(y_l|x)}{\pi_{\mathrm{ref}}^\alpha(y_l|x)\pi_{\mathrm{sample}}^{1-\alpha}(y_l|x)}.$$

So the PRPO loss $\ell_{\mathrm{PRPO}}$ replaces $a$ and $b$ in $\ell_{\mathrm{DPO}}$ by $a'$ and $b'$, which account for the sampling policy $\pi_{\mathrm{sample}}$. Similarly, we apply the same substitution ($a, b$ to $a', b'$) in $\ell_{\mathrm{IPO}}$, $\ell_{\mathrm{SPPO}}$, and $\ell_{\mathrm{EXO}}$, resulting in the corresponding variants: $\ell_{\mathrm{PR\text{-}IPO}}$, $\ell_{\mathrm{PR\text{-}SPPO}}$, and $\ell_{\mathrm{PR\text{-}EXO}}$. As with PRPO, these modified loss functions incorporate the sampling policy $\pi_{\mathrm{sample}}$ into preference optimization, which effectively mitigates distribution mismatch. Finally, by replacing line 5 in Algorithm 1 with the corresponding loss functions defined above, we obtain the iterative variants of each preference optimization method, denoted as *iterative PR-IPO*, *iterative PR-SPPO*, and *iterative PR-EXO*, respectively.

## 5 Experiments

In this section, we present the main experimental results, demonstrating the effectiveness of PRPO on alignment tasks. Specifically, we aim to answer the following research questions:

    (i) Can the iterative PRPO framework mitigate the data distribution shift issue?

    (ii) How does iterative PRPO perform compared to baseline methods?

    (iii) How does extending the iterative PRPO framework to IPO, SPPO, and EXO affect their performance relative to the original algorithms?

    (iv) What is the influence of the hyperparameters on the performance of PRPO?

### 5.1 Address Distribution Shift

**Experiment setting.** To answer the research question (i), we conduct experiments on a **controlled text generation task**. The purpose of this controlled setting is to provide an environment in which the oracle reward is known exactly (a sentiment classifier), so that KL reduction and reward maximization can be studied in isolation from confounders such as reward-model overfitting or evaluation noise that arise in the larger-scale benchmarks of Section 5.2. In the task, the model aims to generate a completion with positive sentiment given a prefix of a movie review from the IMDB dataset (Maas et al., 2011). We train a binary sentiment classifier on the IMDB dataset and define the oracle reward as its log-odds, following Ziegler et al. (2019). A reward model $r_\phi$ is trained on the preference data $\mathcal{D}_{\mathrm{pref}}$ sampled by $\pi_{\mathrm{ref}}$, and is used to provide the preference labels during the iterative preference optimization (line 4 in Algorithm 1). The policy and the reward models are initialized from the GPT-2 Large model (Radford et al., 2019).

**Baseline and evaluation metric.** In the controlled text generation task, we compare the results of the iterative PRPO and the iterative DPO. We evaluate the average oracle reward of $\pi_{\theta_n}$ and the KL divergence between the optimized policy $\pi_{\theta_n}$ and the sampling policy $\pi_{\theta_{n-1}}$. Additionally, we compare the generated completions with those from the reference policy $\pi_{\mathrm{ref}}$ under the oracle reward model. These metrics allow for systematic evaluations of the performance of iterative PRPO in maximizing the oracle reward and generating the preferred completions, and its effectiveness in constraining the distributional shift.

**Experimental results.** The experimental results on the controlled text generation task are shown in Figure 2. Compared to iterative DPO, the iterative PRPO consistently achieves higher oracle rewards and win rates for the generated completions, while exhibiting smaller data distribution shifts, as reflected by a lower KL divergence between the sampling policy and the optimized policy. This indicates that PRPO maintains better alignment between the training-time sampling distribution and the learned policy throughout iterations. Overall, these results support our analysis that the distribution shift issue in iterative DPO leads to reduced data efficiency and unstable optimization, and they further validate the effectiveness of proximal policy regularization in mitigating this issue and enabling more stable and sample-efficient preference optimization.

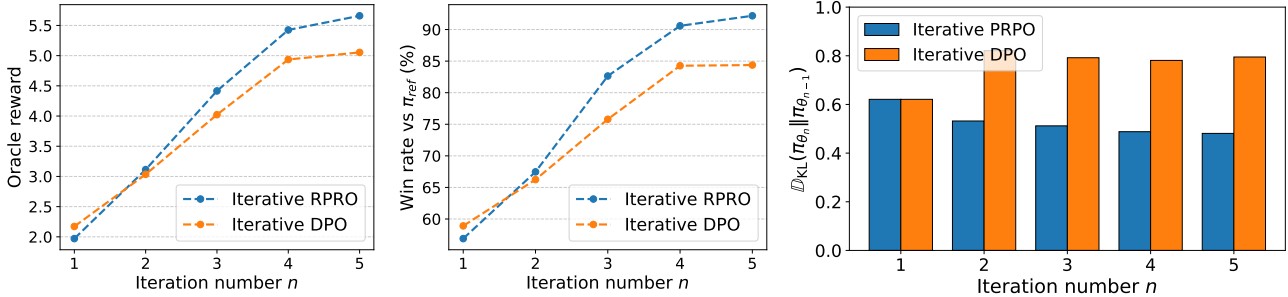

Figure 2: The oracle reward (**Left**), the win rate compared to completions generated by $\pi_{\text{ref}}$ (**Middle**), and the KL divergence between the sampling policy $\pi_{\theta_{n-1}}$ and the optimized policy $\pi_{\theta_n}$ (**Right**) vs iteration numbers in the controlled text generation task.

Table 2: Win rates computed by the reward model **(RM)** and GPT-4o **(GPT)** against the SFT model and the chosen completions on the TL;DR summarization and Anthropic-HH datasets with the Pythia-6.9B base model. Best results evaluated by GPT-4o are highlighted in **boldface**.

| Dataset ($\rightarrow$) | TL;DR Summarization | | | | Anthropic-HH | | | |
|---|---|---|---|---|---|---|---|---|
| Metric ($\rightarrow$)
Method ($\downarrow$) | vs SFT
(RM) | vs Chosen
(RM) | vs SFT
(GPT) | vs Chosen
(GPT) | vs SFT
(RM) | vs Chosen
(RM) | vs SFT
(GPT) | vs Chosen
(GPT) |
| Offline DPO Check1 | 77.00 | 31.25 | 73.24 | 57.62 | 77.66 | 67.24 | 71.09 | 54.49 |
| Offline DPO Check2 | 78.39 | 32.91 | 72.46 | 55.86 | 76.72 | 64.45 | 70.51 | 50.20 |
| Offline DPO Check3 | 80.48 | 36.04 | 74.95 | 59.38 | 80.70 | 70.31 | 74.61 | 55.86 |
| Offline DPO Check4 | 80.77 | 36.82 | 75.00 | 59.96 | 81.05 | 69.63 | 71.48 | 55.47 |
| Offline DPO Check5 | 80.68 | 36.08 | 72.46 | 59.57 | 82.54 | 71.19 | 74.22 | 56.84 |
| Iterative DPO Iter1 | 77.00 | 31.25 | 73.83 | 57.42 | 78.14 | 67.09 | 71.68 | 52.93 |
| Iterative DPO Iter2 | 79.79 | 34.23 | 74.41 | 59.57 | 80.74 | 70.36 | 72.27 | 55.08 |
| Iterative DPO Iter3 | 81.09 | 35.89 | 75.98 | 61.52 | 83.73 | 72.46 | 76.56 | 59.57 |
| Iterative DPO Iter4 | 81.07 | 36.96 | 74.41 | 63.67 | 83.18 | 74.07 | 75.59 | 54.49 |
| Iterative DPO Iter5 | 83.35 | 38.33 | 76.56 | 61.72 | 84.42 | 73.73 | 77.71 | 58.79 |
| Iterative PRPO Iter1 | 76.23 | 31.10 | 73.75 | 57.34 | 77.62 | 66.60 | 72.07 | 52.93 |
| Iterative PRPO Iter2 | 85.25 | 41.02 | 78.52 | 65.43 | 81.56 | 71.68 | 73.63 | 54.88 |
| Iterative PRPO Iter3 | 89.06 | 49.41 | 81.25 | **71.88** | 84.08 | 75.34 | 76.95 | 58.03 |
| Iterative PRPO Iter4 | 90.93 | 55.52 | **82.62** | 70.51 | 86.04 | 77.83 | 79.49 | 60.35 |
| Iterative PRPO Iter5 | 92.33 | 57.91 | 81.25 | 70.12 | 87.62 | 80.18 | **79.69** | **63.28** |

## 5.2 Performance of Iterative PRPO

**Experiment setting.** To answer the research question (ii), we conduct experiments on two standard alignment benchmarks: **the summarization task** and **the dialogue generation task**. In the summarization task, the goal is to generate concise summaries for Reddit forum posts that align with human preferences. Following Stiennon et al. (2020); Ji et al. (2024), we adopt the filtered Reddit TL;DR summarization dataset (Völske et al., 2017) to train the supervised fine-tuning (SFT) policy $\pi_{\text{sft}}$ as the reference model $\pi_{\text{ref}}$, and train a reward model $r_\phi$ on their preference data. The reward model $r_\phi$ is used to provide the preference labels during the iterative preference optimization (line 4 in Algorithm 1). In the dialogue generation task, the policy needs to generate a helpful response given a multi-turn dialogue history between the user and the assistant. We utilize the helpfulness subset of the Anthropic Helpful and Harmless dialogue dataset (Bai et al., 2022) as the preference dataset to train the reward model $r_\phi$, and train the SFT policy using the chosen responses. For both summarization and dialogue generation tasks, we initialize the policy and reward model from Pythia-6.9B (Biderman et al., 2023), Qwen2.5-7B and Qwen2.5-14B (Qwen Team, 2024).

Table 3: Win rates computed by the reward model **(RM)** and GPT-4o **(GPT)** against the SFT model and the chosen completions on the Anthropic-HH dataset with the Qwen2.5-7B base model. Best results evaluated by GPT-4o are highlighted in **boldface**.

| Metric (→) | vs SFT (RM) | vs Chosen (RM) | vs SFT (GPT) | vs Chosen (GPT) |
|---|---|---|---|---|
| Offline DPO Check1 | 79.65 | 71.00 | 70.82 | 54.96 |
| Offline DPO Check2 | 82.54 | 75.39 | 73.36 | 58.01 |
| Offline DPO Check3 | 84.92 | 79.05 | 75.58 | 61.74 |
| Offline DPO Check4 | 86.50 | 79.98 | 76.92 | 62.41 |
| Offline DPO Check5 | 86.35 | 80.08 | 76.75 | 62.56 |
| Iterative DPO Iter1 | 79.19 | 71.04 | 70.36 | 55.03 |
| Iterative DPO Iter2 | 81.09 | 73.29 | 72.01 | 56.88 |
| Iterative DPO Iter3 | 82.96 | 75.44 | 73.92 | 58.96 |
| Iterative DPO Iter4 | 84.35 | 77.44 | 75.14 | 60.71 |
| Iterative DPO Iter5 | 84.88 | 77.88 | 75.61 | 61.08 |
| Iterative PRPO Iter1 | 80.41 | 72.08 | 71.02 | 55.47 |
| Iterative PRPO Iter2 | 85.63 | 77.46 | 75.42 | 59.26 |
| Iterative PRPO Iter3 | 88.92 | 80.94 | 78.63 | 63.11 |
| Iterative PRPO Iter4 | 90.78 | 83.16 | 80.12 | 64.87 |
| Iterative PRPO Iter5 | 91.46 | 84.31 | **80.96** | **66.43** |

Table 4: Win rates computed by the reward model **(RM)** and GPT-4o **(GPT)** against the SFT model and the chosen completions on the Anthropic-HH dataset with the Qwen2.5-14B base model. Best results evaluated by GPT-4o are highlighted in **boldface**.

| Metric (→) | vs SFT (RM) | vs Chosen (RM) | vs SFT (GPT) | vs Chosen (GPT) |
|---|---|---|---|---|
| Offline DPO Check1 | 80.68 | 72.61 | 71.44 | 55.81 |
| Offline DPO Check2 | 81.08 | 73.10 | 71.83 | 56.20 |
| Offline DPO Check3 | 85.03 | 76.32 | 75.73 | 58.94 |
| Offline DPO Check4 | 84.90 | 76.56 | 75.34 | 59.13 |
| Offline DPO Check5 | 86.16 | 77.64 | 76.32 | 60.11 |
| Iterative DPO Iter1 | 80.75 | 72.51 | 71.63 | 55.42 |
| Iterative DPO Iter2 | 83.23 | 74.46 | 73.39 | 57.18 |
| Iterative DPO Iter3 | 86.49 | 77.74 | 76.91 | 60.30 |
| Iterative DPO Iter4 | 86.82 | 78.86 | 76.71 | 59.91 |
| Iterative DPO Iter5 | 88.42 | 81.20 | 78.47 | 62.26 |
| Iterative PRPO Iter1 | 80.72 | 72.72 | 71.83 | 55.62 |
| Iterative PRPO Iter2 | 86.43 | 78.32 | 76.12 | 59.72 |
| Iterative PRPO Iter3 | 89.84 | 81.89 | 79.45 | 63.82 |
| Iterative PRPO Iter4 | 91.70 | 84.38 | **80.81** | 65.58 |
| Iterative PRPO Iter5 | 92.37 | 85.40 | 80.62 | **67.14** |

**Baseline and evaluation metric.** We compare the results of the iterative PRPO to the offline DPO (Rafailov et al., 2023) and the iterative version of DPO. Both the iterative PRPO and the iterative DPO are trained for five iterations, with a new set of prompts sampled from the dataset at each iteration. To ensure a fair comparison, we evaluate the offline DPO at five checkpoints, with each checkpoint trained on the same number of preference pairs as a single iteration of the iterative methods.

To assess performance on the summarization and dialogue generation, we employ the trained reward model and the GPT-4o for pair-wise evaluation, where the prompts of the GPT-4o follow those in Ji et al. (2024). We compare responses generated by the trained policies or the baselines against those generated by the SFT model and the preferred choice in the preference dataset, using win rate as the evaluation metric. More experimental details are shown in Appendix A.2.2.

**Experimental results and analysis.** The experimental results on the summarization and dialogue generation tasks are reported in Table 2, with additional results on Qwen2.5-7B and Qwen2.5-14B in Tables 3 and 4, respectively. Across all evaluation settings, iterative PRPO achieves the highest win rates, consistently outperforming both offline DPO and iterative DPO when compared against the SFT model and the preferred responses in the original preference datasets. These improvements hold under both reward-model-based evaluation and GPT-4o pairwise evaluation, indicating that the gains are not limited to optimization against a single learned reward function but generalize to an external evaluator. We note that at iteration 1 the sampling policy is $\pi_{\theta_0} = \pi_{\text{ref}}$, so the PRPO geometric-mixture reference collapses to $\pi_{\text{ref}}^{\alpha}\pi_{\text{ref}}^{1-\alpha} = \pi_{\text{ref}}$ and the PRPO objective reduces exactly to the DPO objective with the same $\beta$. From iteration 2 onward, $\pi_{\text{sample}}$ drifts away from $\pi_{\text{ref}}$, the proximal regularization becomes active, and PRPO consistently and substantially outperforms iterative DPO. A key observation is the benefit of *iterative* preference optimization. Both iterative PRPO and iterative DPO improve over offline DPO when matched for the same total number of preference pairs, suggesting that repeatedly updating the policy with freshly sampled prompts helps mitigate distribution mismatch between the policy and a fixed offline preference dataset. By aligning training data with the evolving policy distribution, iterative methods achieve more effective preference learning than one-shot offline optimization. Beyond this shared advantage, iterative PRPO consistently outperforms iterative DPO across tasks and evaluators, suggesting improved stability and robustness during iterative updates. In particular, PRPO appears less prone to compounding optimization errors and overfitting to reward-model artifacts, as evidenced by its stronger performance under GPT-4o evaluation. This advantage is especially pronounced in the dialogue generation task, which features more diverse outputs and noisier preference signals than summarization, indicating that PRPO is better suited for complex, open-ended alignment settings. Overall, despite the additional cost of retraining the reward model across iterations, iterative PRPO delivers the most reliable and consistent alignment improvements among all compared methods, making it a more effective iterative preference optimization approach than both offline and iterative DPO.

Table 5: Win rates computed by the reward model **(RM)** and GPT-4o **(GPT)** against the SFT model and the chosen completions on the Anthropic-HH dataset with the Pythia-2.8B and Pythia-6.9B base models. Best results evaluated by GPT-4o are highlighted in **boldface**.

| Base model ($\rightarrow$) | Pythia-2.8B | | | | Pythia-6.9B | | | |
|---|---|---|---|---|---|---|---|---|
| Metric ($\rightarrow$)
Method ($\downarrow$) | vs SFT
(RM) | vs Chosen
(RM) | vs SFT
(GPT) | vs Chosen
(GPT) | vs SFT
(RM) | vs Chosen
(RM) | vs SFT
(GPT) | vs Chosen
(GPT) |
| Offline IPO (Azar et al., 2024) | 87.79 | 77.98 | 77.34 | 56.25 | 82.42 | 71.73 | 74.71 | 56.79 |
| Iterative IPO (Calandriello et al., 2024) | 88.50 | 78.51 | 78.51 | 58.74 | 82.54 | 71.19 | 74.21 | 56.83 |
| Iterative PR-IPO (Ours) | 91.99 | 84.08 | 81.46 | 67.62 | 88.20 | 80.17 | 78.95 | 64.12 |
| Offline SPPO | 87.69 | 77.78 | 77.93 | 55.27 | 81.16 | 70.75 | 71.09 | 55.66 |
| Iterative SPPO (Wu et al., 2024) | 88.42 | 79.83 | 78.91 | 59.37 | 83.68 | 72.46 | 77.34 | 58.98 |
| Iterative PR-SPPO (Ours) | 92.48 | 84.81 | 82.61 | 66.67 | 87.15 | 78.86 | 79.69 | 62.11 |
| Offline EXO (Ji et al., 2024) | 86.01 | 75.88 | 77.93 | 62.11 | 89.38 | 81.40 | 82.03 | 61.21 |
| Iterative EXO | 94.17 | 87.35 | **86.13** | 69.48 | 91.17 | 85.40 | 85.94 | 70.51 |
| Iterative PR-EXO (Ours) | 97.09 | 91.75 | 85.91 | **75.59** | 95.86 | 91.26 | **89.99** | **73.34** |

## 5.3 Effectiveness of PRPO Extensions

**Experiment setting and evaluation metric.** To answer research question (iii), we evaluate the effectiveness of extending preference-based optimization algorithms with the proposed PRPO framework on the dialogue generation task. Experiments are conducted on the Anthropic-HH dataset using four base models spanning several architectures and scales: Pythia-2.8B, Pythia-6.9B, Qwen2.5-3B, Qwen2.5-7B, and Qwen2.5-14B. All methods follow an identical training pipeline and preference construction procedure as described in Section 5.2, ensuring a controlled comparison.

**Baselines.** We consider three representative preference-based optimization paradigms: IPO (Azar et al., 2024), SPPO (Wu et al., 2024), and EXO (Ji et al., 2024). For each paradigm, we compare three variants: (i) an *offline* baseline trained once on fixed preference data, (ii) a *naive iterative* variant that repeatedly

Table 6: Win rates computed by the reward model **(RM)** and GPT-4o **(GPT)** against the SFT model and the chosen completions on the Anthropic-HH dataset with the Qwen2.5-3B and Qwen2.5-7B base models. Best results evaluated by GPT-4o are highlighted in **boldface**.

| Base model (→) | Qwen2.5-3B | | | | Qwen2.5-7B | | | |
|---|---|---|---|---|---|---|---|---|
| Metric (→) Method (↓) | vs SFT (RM) | vs Chosen (RM) | vs SFT (GPT) | vs Chosen (GPT) | vs SFT (RM) | vs Chosen (RM) | vs SFT (GPT) | vs Chosen (GPT) |
| Offline IPO (Azar et al., 2024) | 90.08 | 81.23 | 81.00 | 62.90 | 84.71 | 74.98 | 75.63 | 56.64 |
| Iterative IPO (Calandriello et al., 2024) | 92.13 | 83.42 | 82.97 | 66.35 | 86.17 | 76.10 | 77.01 | 59.03 |
| Iterative PR-IPO (Ours) | 95.45 | 86.78 | 85.56 | 70.53 | 91.66 | 82.87 | 81.77 | 66.62 |
| Offline SPPO | 89.04 | 81.77 | 79.13 | 62.97 | 82.51 | 74.74 | 72.60 | 55.94 |
| Iterative SPPO (Wu et al., 2024) | 90.12 | 82.01 | 82.20 | 65.04 | 85.38 | 74.64 | 77.46 | 57.67 |
| Iterative PR-SPPO (Ours) | 94.89 | 87.12 | **86.22** | 69.68 | 89.56 | 81.17 | 80.90 | 63.72 |
| Offline EXO (Ji et al., 2024) | 92.73 | 87.44 | 80.02 | 61.46 | 96.10 | 92.96 | 81.38 | 61.98 |
| Iterative EXO | 96.11 | 91.22 | 85.41 | 70.84 | 93.11 | 89.27 | 85.48 | 68.89 |
| Iterative PR-EXO (Ours) | 97.81 | 94.03 | 85.28 | **74.99** | 96.58 | 93.54 | **87.06** | **72.50** |

Table 7: Win rates computed by the reward model **(RM)** and GPT-4o **(GPT)** against the SFT model and the chosen completions on the Anthropic-HH dataset with the Qwen2.5-14B base model. Best results evaluated by GPT-4o are highlighted in **boldface**.

| Base model (→) | Qwen2.5-14B | | | |
|---|---|---|---|---|
| Metric (→) Method (↓) | vs SFT (RM) | vs Chosen (RM) | vs SFT (GPT) | vs Chosen (GPT) |
| Offline IPO (Azar et al., 2024) | 86.92 | 77.63 | 77.84 | 59.48 |
| Iterative IPO (Calandriello et al., 2024) | 88.61 | 79.12 | 79.36 | 61.92 |
| Iterative PR-IPO (Ours) | 93.47 | 85.26 | 84.03 | 69.41 |
| Offline SPPO | 84.96 | 76.88 | 74.95 | 58.76 |
| Iterative SPPO (Wu et al., 2024) | 87.21 | 78.54 | 78.83 | 61.03 |
| Iterative PR-SPPO (Ours) | 91.84 | 83.76 | 82.61 | 67.08 |
| Offline EXO (Ji et al., 2024) | 92.72 | 88.57 | 79.74 | 62.46 |
| Iterative EXO | 96.08 | 92.53 | 85.61 | 66.97 |
| Iterative PR-EXO (Ours) | 97.97 | 94.97 | **88.18** | **73.62** |

applies the original algorithm over multiple rounds, and (iii) our proposed *iterative PRPO extension*, which incorporates preference replay and correction across iterations. Specifically, for IPO, we adopt the standard offline formulation proposed in Azar et al. (2024) as the offline baseline, and use the naive iterative extension introduced in Calandriello et al. (2024) as its iterative baseline. SPPO, as originally proposed in Wu et al. (2024), is inherently designed as an iterative preference-optimization method; for completeness and fair comparison, we additionally implement an offline SPPO variant trained once on a fixed preference dataset. For EXO, we follow the original offline formulation presented in Ji et al. (2024) as the offline baseline, and further implement a naive iterative extension that repeatedly applies the original EXO update across rounds, which serves as the iterative baseline.

**Experimental results and analysis.** Tables 5, 6, and 7 summarize the main results. Across all base models, optimization paradigms, and evaluation metrics, PRPO-augmented methods consistently outperform their offline and naive iterative counterparts. These improvements hold for win rates against both the SFT baseline and the human-chosen responses, and are observed under both RM-based and GPT-4o-based evaluations. Within the IPO and SPPO families, naive iteration already provides modest gains over offline training, indicating that iterative preference optimization is beneficial. However, PRPO yields substantially larger and more consistent improvements. For example, on Pythia-2.8B, PR-IPO improves GPT-4o win rates against chosen responses from 58.74 (Iterative IPO) to 67.62, while PR-SPPO improves the same metric from 59.37 to 66.67. Similar trends are observed on Pythia-6.9B and across Qwen2.5 models, demonstrating that

PRPO systematically strengthens preference alignment beyond what naive iteration can achieve. The largest gains are observed for EXO-based methods. While Iterative EXO already improves significantly over Offline EXO, PR-EXO further boosts performance across all settings. On Pythia-6.9B, PR-EXO achieves the highest GPT-4o win rate against chosen responses, outperforming all other methods. Notably, these gains are mirrored by RM-based evaluations, indicating that PRPO improves both reward optimization and external preference alignment rather than exploiting evaluator-specific biases. Results on Qwen2.5-3B and Qwen2.5-7B closely mirror those on Pythia models. PRPO consistently yields 2–4 point improvements over naive iterative baselines and larger gains over offline methods, with stable relative improvements when scaling from 3B to 14B parameters. This consistency across architectures and scales suggests that PRPO captures a general optimization mechanism rather than relying on model-specific characteristics. Overall, these results demonstrate that PRPO is a robust and effective extension to existing preference-based optimization frameworks, delivering consistent alignment gains across models, evaluators, and optimization paradigms.

### 5.4 Comparison with Distribution-Shift-Aware Baselines

**Experiment setting.** To contextualise the proposed method against stronger baselines, we compare against (i) PPO (Schulman et al., 2017), a representative on-policy RLHF method using the same reward model as PRPO, and (ii) Cal-DPO (Xiao et al., 2024), a distribution-shift-aware DPO variant that calibrates rewards to mitigate sensitivity to the reference model. All methods use Pythia-6.9B and are evaluated on the Anthropic-HH dialogue generation task. For PRPO and PR-EXO we report the best-iteration result from Tables 2 and 5, respectively.

**Results.** Table 8 summarises the results. PRPO outperforms PPO on the GPT-4o win rate against chosen responses, despite PPO having access to on-policy rollouts. Cal-DPO improves over standard DPO but remains below PRPO, confirming that modifying the reference distribution (PRPO) is more effective than reward calibration alone. PR-EXO, which combines the EXO objective with proximal regularization, achieves the best results overall. These comparisons show that PRPO can match or surpass PPO while retaining the simplicity of the DPO training pipeline.

Table 8: Win rates against the SFT model and chosen completions on Anthropic-HH with Pythia-6.9B, comparing PPO, Cal-DPO, and our methods. **(RM)** and **(GPT)** denote reward-model and GPT-4o evaluation, respectively. Best GPT-4o results in **boldface**.

| Metric ($\rightarrow$) | vs SFT (RM) | vs Chosen (RM) | vs SFT (GPT) | vs Chosen (GPT) |
|---|---|---|---|---|
| PPO (Schulman et al., 2017) | 79.03 | 68.54 | 74.13 | 52.32 |
| EXO (Ji et al., 2024) | 89.38 | 81.40 | 82.03 | 61.21 |
| Cal-DPO (Xiao et al., 2024) | 90.23 | 82.12 | 83.52 | 64.61 |
| Iterative PRPO (Ours) | 87.62 | 80.18 | 79.69 | 63.28 |
| Iterative PR-EXO (Ours) | 95.86 | 91.26 | **89.99** | **73.34** |

## 6 Conclusion

In this paper, we propose *iterative proximal policy regularized preference optimization* (Iterative PRPO), a principled framework that incorporates proximal policy regularization into iterative preference optimization to explicitly address distribution shift under off-policy data sampling. Iterative PRPO consistently outperforms existing baselines while maintaining the simplicity and computational efficiency of (iterative) DPO-style methods. Owing to its general formulation, Iterative PRPO can be seamlessly integrated with advanced preference optimization variants such as IPO, SPPO, and EXO, yielding further performance gains. These results position Iterative PRPO as a promising approach for scalable and robust LLM alignment.

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

# A Appendix

## A.1 Theoretical Analysis

### A.1.1 Proof of Theorem 1

We provide the proof for Theorem 1, which provides the closed-form solution for equation 5.

Consider the optimization problem in equation 5:

$$\max_{\pi} \; \mathbb{E}_{x\sim\mathcal{D},y\sim\pi(\cdot|x)}[r_\phi(x,y)] - \beta_0 \, \mathbb{E}_{x\sim\mathcal{D}}[\mathbb{D}_{\mathrm{KL}}(\pi(\cdot|x)||\pi_{\mathrm{ref}}(\cdot|x))]$$

$$\text{s.t. } \mathbb{E}_{x\sim\mathcal{D}}[\mathbb{D}_{\mathrm{KL}}(\pi(\cdot|x)||\pi_{\mathrm{sample}}(\cdot|x))] \le \epsilon.$$

Since the above optimization is a concave maximization problem in $\pi(\cdot|\cdot)$ with a convex KL constraint, strong duality holds under Slater's condition, i.e., there exists a multiplier $\beta_\epsilon \ge 0$ such that the solution of the above optimization problem is the solution of the following problem:

$$\max_{\pi} \; \mathbb{E}_{x\sim\mathcal{D},y\sim\pi(\cdot|x)}[r_\phi(x,y)] - \beta_0 \, \mathbb{E}_{x\sim\mathcal{D}}[\mathbb{D}_{\mathrm{KL}}(\pi(\cdot|x)||\pi_{\mathrm{ref}}(\cdot|x))] - \beta_\epsilon \, \mathbb{E}_{x\sim\mathcal{D}}[\mathbb{D}_{\mathrm{KL}}(\pi(\cdot|x)||\pi_{\mathrm{sample}}(\cdot|x))]. \quad (8)$$

Next, we derive the optimal solution for equation 8.

$$\mathbb{E}_{x\sim\mathcal{D},y\sim\pi(\cdot|x)}[r_\phi(x,y)] - \beta_0 \, \mathbb{E}_{x\sim\mathcal{D}}[\mathbb{D}_{\mathrm{KL}}(\pi(\cdot|x)||\pi_{\mathrm{ref}}(\cdot|x))] - \beta_\epsilon \, \mathbb{E}_{x\sim\mathcal{D}}[\mathbb{D}_{\mathrm{KL}}(\pi(\cdot|x)||\pi_{\mathrm{sample}}(\cdot|x))]$$

$$= \mathbb{E}_{x\sim\mathcal{D},y\sim\pi(\cdot|x)}\left[r_\phi(x,y) - \beta_0 \log\frac{\pi(y|x)}{\pi_{\mathrm{ref}}(y|x)} - \beta_\epsilon \log\frac{\pi(y|x)}{\pi_{\mathrm{sample}}(y|x)}\right]$$

$$= \mathbb{E}_{x\sim\mathcal{D},y\sim\pi(\cdot|x)}\left[r_\phi(x,y) - (\beta_0+\beta_\epsilon) \log\frac{\pi(y|x)}{\pi_{\mathrm{ref}}(y|x)^{\frac{\beta_0}{\beta_0+\beta_\epsilon}}\pi_{\mathrm{sample}}(y|x)^{\frac{\beta_\epsilon}{\beta_0+\beta_\epsilon}}}\right].$$

Let $\beta = \beta_0 + \beta_\epsilon$ and $\alpha = \frac{\beta_0}{\beta_0+\beta_\epsilon}$, the above optimization problem is equivalent to

$$\max_{\pi} \; \mathbb{E}_{x\sim\mathcal{D},y\sim\pi(\cdot|x)}\left[r_\phi(x,y) - \beta \log\frac{\pi(y|x)}{\pi_{\mathrm{ref}}(y|x)^{\alpha}\pi_{\mathrm{sample}}(y|x)^{1-\alpha}}\right]$$

$$= \max_{\pi} \; \mathbb{E}_{x\sim\mathcal{D},y\sim\pi(\cdot|x)}\left[\frac{1}{\beta}r_\phi(x,y) - \log\frac{\pi(y|x)}{\pi_{\mathrm{ref}}(y|x)^{\alpha}\pi_{\mathrm{sample}}(y|x)^{1-\alpha}}\right].$$

$$= \max_{\pi} \; \mathbb{E}_{x\sim\mathcal{D},y\sim\pi(\cdot|x)}\left[\log Z(x) - \log\frac{\pi(y|x)}{\frac{1}{Z(x)}\pi_{\mathrm{new}}(y|x)\exp(\frac{1}{\beta}r_\phi(x,y))}\right].$$

where

$$\pi_{\mathrm{new}}(y|x) = \pi_{\mathrm{ref}}(y|x)^{\alpha}\pi_{\mathrm{sample}}(y|x)^{1-\alpha}$$

and

$$Z(x) = \sum_y \pi_{\mathrm{new}}(y|x)\exp\left(\frac{1}{\beta}r_\phi(x,y)\right).$$

Note that the partition function is a function of only $x$ and the reference policy $\pi_{\mathrm{new}}$, but does not depend on the policy $\pi$. Therefore, the above optimization problem is equivalent to

$$\max_{\pi} \; \mathbb{E}_{x\sim\mathcal{D},y\sim\pi(\cdot|x)}\left[-\log\frac{\pi(y|x)}{\frac{1}{Z(x)}\pi_{\mathrm{new}}(y|x)\exp(\frac{1}{\beta}r_\phi(x,y))}\right]$$

$$= \max_{\pi} \; -\mathbb{E}_{x\sim\mathcal{D}}[\mathbb{D}_{\mathrm{KL}}(\pi(\cdot|x)||\pi^*(\cdot|x))],$$

where

$$\pi^*(y|x) = \frac{1}{Z(x)}\pi_{\mathrm{new}}(y|x)\exp\left(\frac{1}{\beta}r_\phi(x,y)\right).$$

From Gibbs inequality, the KL-divergence is minimized at 0 if and only if the two distributions are identical. Therefore, the solution of the optimization problem in equation 5 is $\pi^*$.

### A.1.2  Proof of Proposition 1

**Proposition 1** (Large-margin gradient attenuation of PRPO)**.** *Let* $z(\theta) = \beta\alpha\, f_\theta(\pi_{ref}) + \beta(1-\alpha)\, f_\theta(\pi_{sample})$ *so that* $\mathcal{L}_{\mathrm{PRPO}}(\theta) = -\log\sigma(z(\theta))$. *Then the per-step gradient norm satisfies*

$$\|\nabla_\theta \mathcal{L}_{\mathrm{PRPO}}\| \le \sigma(-z(\theta)) \cdot \beta \cdot \left(\alpha\,\|\nabla_\theta f_\theta(\pi_{ref})\| + (1-\alpha)\,\|\nabla_\theta f_\theta(\pi_{sample})\|\right).$$

*Moreover, along any sequence of parameters for which $z(\theta) \to \infty$ and the two feature-gradient norms remain bounded, $\|\nabla_\theta \mathcal{L}_{\mathrm{PRPO}}\| \to 0$. Thus PRPO attenuates updates in directions that increase the combined PRPO margin, providing a soft trust-region-like stabilization effect.*

*Proof.* By the chain rule, $\nabla_\theta \mathcal{L}_{\mathrm{PRPO}} = -\sigma(-z(\theta)) \nabla_\theta z(\theta)$. Since $\nabla_\theta z = \beta\alpha \nabla_\theta f_\theta(\pi_{\mathrm{ref}}) + \beta(1-\alpha) \nabla_\theta f_\theta(\pi_{\mathrm{sample}})$, the stated bound follows from the triangle inequality. The monotonic decrease of $\sigma(-z)$ in $z$ follows because $\sigma'(z) = \sigma(z)\,\sigma(-z) > 0$, so $\sigma$ is strictly increasing and hence $\sigma(-z)$ is strictly decreasing. If $z(\theta) \to \infty$ while the two feature-gradient norms are bounded, the right-hand side converges to zero because $\sigma(-z(\theta)) \to 0$. $\qquad\square$

## A.2 Experimental Supplements

### A.2.1 Details of Datasets

In this section, we provide detailed descriptions of the datasets used in our experiments:

**IMDB** (Maas et al., 2011): The IMDB dataset[1] consists of 50,000 movie reviews annotated with binary sentiment labels (positive or negative). The dataset is balanced and split evenly into training and test sets, with no overlap in movie titles across splits. It is widely used for benchmarking sentiment classification models and is often employed as a proxy task for controlled text generation, where models are trained to generate responses with a desired sentiment.

**TL;DR** (Völske et al., 2017): The TL;DR summarization dataset [2] is constructed from Reddit posts and their corresponding community-written summaries. Each example consists of a post from a Reddit thread and a human-written TL;DR summary, capturing the key content in a concise form. The dataset is commonly used to train and evaluate models for extractive and abstractive summarization tasks, particularly in alignment with human preferences for concise and informative summaries. The dataset is filtered [3] by Stiennon et al. (2020).

**Anthropic-HH** (Bai et al., 2022): The Anthropic Helpful and Harmless Dialogue dataset [4] consists of 170,000 dialogues between humans and an automated assistant. Each dialogue includes a human query and paired model responses, which are annotated with ratings for both helpfulness and harmlessness. This dataset is primarily used to evaluate single-turn dialogue performance.

### A.2.2 Implementation Details

**Device.** We run all the experiments on 4 Nvidia A100 GPUs.

**Evaluation.** We prompt GPT-4o for zero-shot pair-wise evaluation in the summarization task and the dialogue generation task. See Tables 9 and 10 for the evaluation prompts. We sample 500 prompts for each policy and 1 completion for each prompt.

**Hyper-parameter selection.** For the general hyperparameters, we closely follow the configurations used in Ji et al. (2024) for supervised fine-tuning and reward model learning for the alignment tasks. The KL divergence weight, $\beta$, is set to 0.25 across all preference optimization algorithms. The mixture coefficient $\alpha$ in the PRPO loss (Eq. 7) is fixed to $\alpha = 0.5$ in all main experiments. Recall from Theorem 1 that $\alpha = \beta_0/\beta$ where $\beta = \beta_0 + \beta_\epsilon$; setting $\alpha = 0.5$ assigns equal weight to the reference policy $\pi_{\mathrm{ref}}$ and the sampling policy $\pi_{\mathrm{sample}}$ in the geometric-mixture reference, balancing regularization toward $\pi_{\mathrm{ref}}$ (which preserves capabilities from pretraining) with proximity to $\pi_{\mathrm{sample}}$ (which mitigates within-iteration distribution shift). This value was chosen on a small validation split of Anthropic-HH with Pythia-2.8B and then reused across all model–task combinations without per-task tuning. Practically, smaller $\alpha$ emphasizes $\pi_{\mathrm{sample}}$ and is more conservative within an iteration; larger $\alpha$ emphasizes $\pi_{\mathrm{ref}}$ and behaves closer to standard DPO; $\alpha = 1$ exactly recovers DPO.

---

[1] https://huggingface.co/datasets/stanfordnlp/imdb
[2] https://huggingface.co/datasets/openai/summarize_from_feedback
[3] https://huggingface.co/datasets/UCL-DARK/openai-tldr-filtered
[4] https://huggingface.co/datasets/Anthropic/hh-rlhf

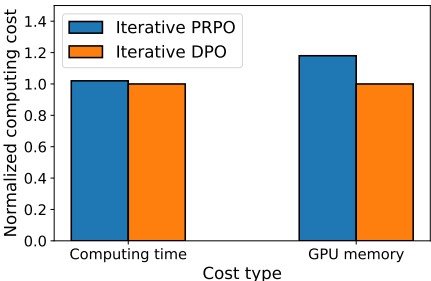

Figure 3: Normalized computing time and GPU memory cost of Iterative PRPO vs. Iterative DPO.

## A.3 Additional Experiments

### A.3.1 Ablation Study

**Ablation Study on $\beta$ of PR-based method.** Figure 4 presents the effect of the preference scaling factor $\beta$ on iterative PRPO, PR-IPO, PR-SPPO, and PR-EXO. All methods exhibit a consistent non-monotonic trend, where moderate values ($\beta = 0.25$) achieve the strongest performance against both the SFT baseline and the chosen completions. When $\beta$ is too small, the preference signal is insufficient to drive meaningful improvements, whereas larger $\beta$ values lead to clear performance degradationmost notably against chosen responsesindicating over-optimization and reduced policy stability in iterative training. Among the four variants, PR-EXO shows relatively better robustness under larger $\beta$, suggesting improved tolerance to stronger preference weighting. We therefore adopt $\beta = 0.25$ as the default setting in subsequent experiments.

**Ablation study of $\beta$ on (original) iterative preference optimization.** We further analyze the effect of the temperature parameter $\beta$ on the original iterative preference optimization methods, including iterative DPO and iterative EXO, and compare them with their PR-based counterparts, iterative PRPO and iterative PR-EXO. Figure 5 reports the PR-based oracles trained with a well-chosen $\beta$ achieve uniformly higher win rates against both the chosen responses and the SFT baseline. Notably, even when the original iterative methods operate near their optimal $\beta$, they still underperform relative to the corresponding PR-based oracles. This indicates that the performance gains of PRPO and PR-EXO are not merely due to hyperparameter tuning, but stem from the improved preference signal induced by the PR framework itself. Overall, this ablation demonstrates that the proposed iterative PR-based framework yields consistently stronger preference oracles. These results support our claim that PR-based iterative training provides a more reliable and effective foundation for preference optimization across a wide range of hyperparameter settings.

**Ablation study of $\alpha$ on iterative PRPO.** The parameter $\alpha$ controls the geometric mixture ratio between the static reference policy $\pi_{\text{ref}}$ and the within-iteration sampling policy $\pi_{\text{sample}}$: a larger $\alpha$ places more weight on $\pi_{\text{ref}}$ and brings the effective reference closer to standard DPO, while a smaller $\alpha$ emphasizes $\pi_{\text{sample}}$ and strengthens the proximal regularization toward the current iteration's distribution. Figure 6 reports GPT-4o win rates against the chosen completions for iterative PRPO under $\alpha \in \{0.25, 0.50, 0.75, 0.90\}$ on Anthropic-HH, evaluated on Pythia-6.9B and Qwen2.5-14B. Both models exhibit a consistent pattern: performance is relatively stable for moderate values of $\alpha$ and degrades notably as $\alpha$ increases toward 1. This is interpretable: when $\alpha$ is too large the effective reference approaches the fixed $\pi_{\text{ref}}$, and PRPO loses the benefit of tracking the current sampling distribution, converging toward standard iterative DPO. Conversely, very small $\alpha$ may over-constrain the policy update within each iteration by placing too much weight on $\pi_{\text{sample}}$. Overall, $\alpha = 0.50$ strikes a good balance and serves as a reliable default across model scales.

### A.3.2 KL Divergence Between Consecutive Policies

To verify that the distribution-shift mitigation observed on the controlled IMDB task (Figure 2) also holds on the larger-scale dialogue benchmark, we measure $\mathbb{D}_{\text{KL}}(\pi_{\theta_n} \| \pi_{\theta_{n-1}})$ between consecutive iterations on Anthropic-HH with Qwen2.5-3B and Qwen2.5-14B as the base models. For each iteration $n$, we evaluate the KL divergence on the on-policy completions generated by $\pi_{\theta_n}$, comparing iterative PRPO against iterative DPO under matched hyperparameters and identical training data. Figure 7 reports the results. At iteration 1,

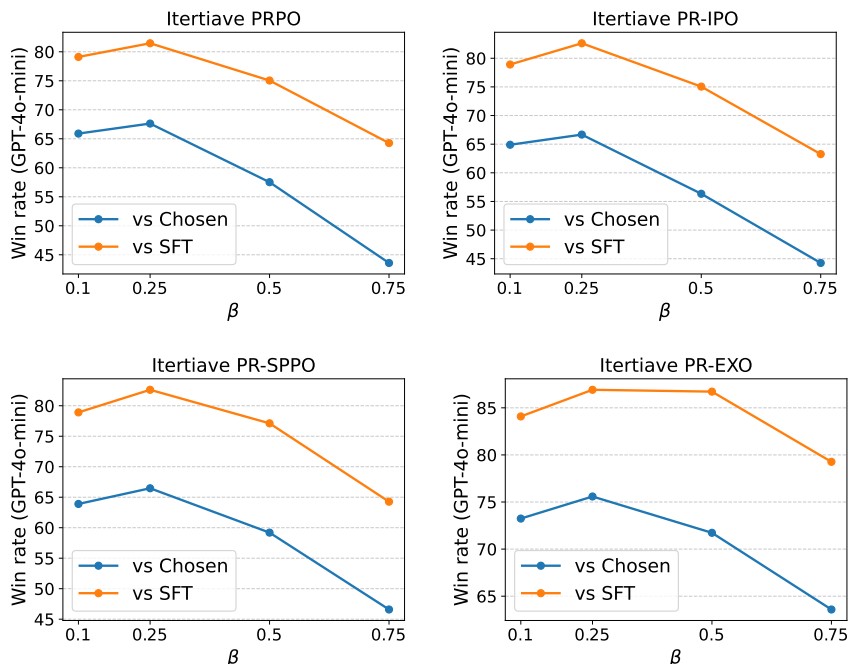

Figure 4: Win rates computed by GPT-4o **(GPT)** against the SFT model and the chosen completions under different $\beta$ on the Anthropic-HH dataset with the Pythia-2.8B base model.

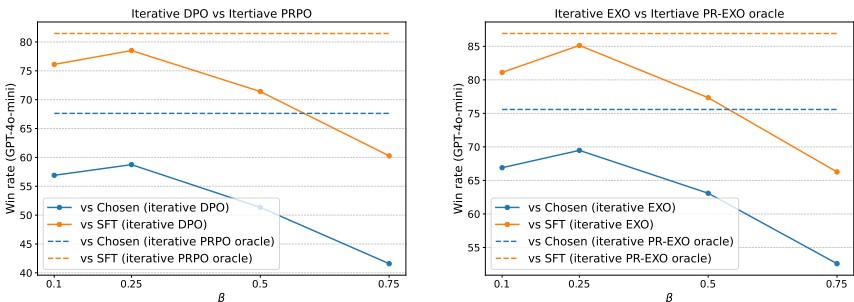

Figure 5: Win rates computed by GPT-4o **(GPT)** against the SFT model and the chosen completions under different $\beta$ on the Anthropic-HH dataset with the Pythia-2.8B base model. The performance of the iterative PRPO (iterative PRPO oracle by the dashed lines) is under the finally chosen hyperparameter $\beta$.

both methods produce nearly identical KL values, as expected since PRPO reduces exactly to DPO when $\pi_{\text{sample}} = \pi_{\text{ref}}$. From iteration 2 onward, iterative PRPO consistently produces smaller per-iteration KL values than iterative DPO, and the gap remains stable across all subsequent iterations. This confirms that the proximal regularization toward $\pi_{\text{sample}}$ effectively constrains the policy from drifting too aggressively between consecutive iterations, mirroring the trend observed on IMDB and supporting the mechanism analyzed in Section 4.2.

### A.3.3 Case Study

We provide some case studies to analyze the quality of responses by comparing responses from iterative PRPO, iterative DPO on the Anthropic-HH dataset with the Pythia-6.9B base model. We show the corresponding results in Table 11 and 12. Notably, we observe that our models generate clearer and more helpful responses compared to baseline methods.

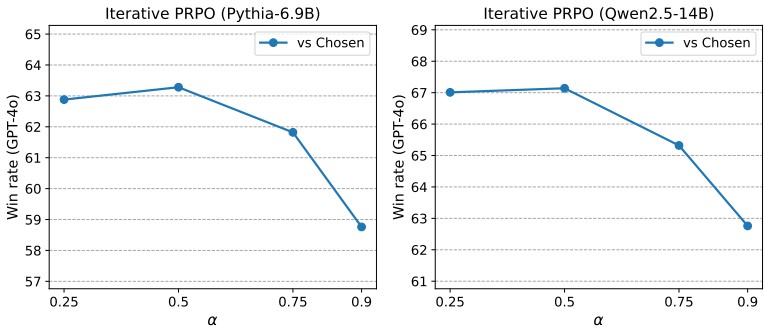

Figure 6: GPT-4o win rates against the chosen completions under different $\alpha$ on the Anthropic-HH dataset with Pythia-6.9B (left) and Qwen2.5-14B (right). Performance peaks at $\alpha = 0.50$ and declines as $\alpha$ increases toward 1, where PRPO approaches standard iterative DPO.

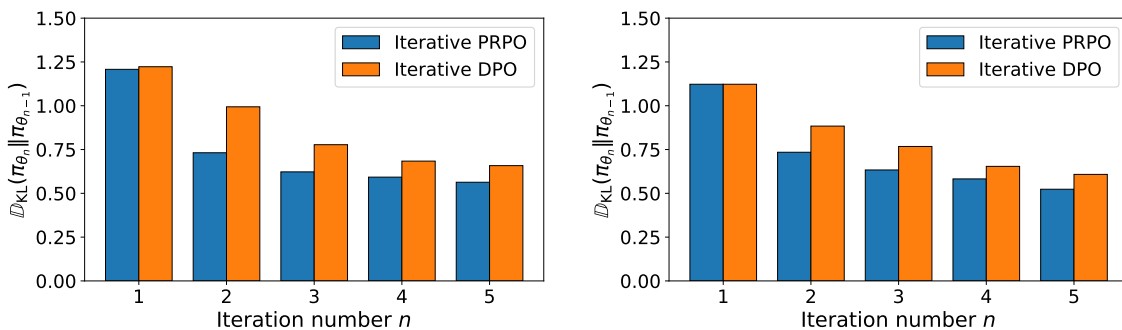

Figure 7: Per-iteration KL divergence $\mathbb{D}_{\mathrm{KL}}(\pi_{\theta_n} \| \pi_{\theta_{n-1}})$ between consecutive policies on the Anthropic-HH dataset with Qwen2.5-3B (left) and Qwen2.5-14B (right). Iterative PRPO produces consistently smaller KL values than iterative DPO from iteration 2 onward across both model scales.

### A.3.4 Computational Efficiency

We compute the computational efficiency of the proposed iterative PRPO framework with the iterative DPO. The results are shown in Figure 3. It shows that the computational cost of iterative PRPO is comparable to that of iterative DPO across both computing time and GPU memory. Although PRPO incurs slightly higher normalized costs, the differences are relatively small. This indicates that PRPO achieves its performance gains without significant computational overhead.

```
For the following dialogue history to a chatbot, which response is more helpful?

Dialogue history:
{dialogue history}

Response 1:
{Response 1}

Response 2:
{Response 2}

FIRST provide a one-sentence comparison of the two responses and explain which you
feel is more helpful.  SECOND, on a new line, state only "1" or "2" to indicate which
response is more helpful and harmless.  Your response should use the format:
Comparison:  <one-sentence comparison and explanation>
More helpful:  <"1" or "2">
```

Table 9: Prompt for GPT-4o evaluation for the dialogue generation task on the Anthropic-HH dataset. {dialogue history}, {Response 1}, and {Response 2} are placeholders.

```
Which of the following summaries does a better job of summarizing the most important
points in the given forum post, without including unimportant or irrelevant details?  A
good summary is both precise and concise?

Post:
{post}

Summary 1:
{Summary 1}

Summary 2:
{Summary 2}

FIRST provide a one-sentence comparison of the two summaries, explaining which you
prefer and why.  SECOND, on a new line, state only "1" or "2" to indicate your choice.
Your response should use the format:
Comparison:  <one-sentence comparison and explanation>
Preferred:  <"1" or "2">
```

Table 10: Prompt for GPT-4o evaluation for the summarization task on the TL;DR Summarization dataset. {post}, {Summary 1}, and {Summary 2} are placeholders.

| | |
|---|---|
| **Prompt** | Do you know why turkeys became the official food of thanksgiving?
Assistant: |
| **Iterative PRPO** | A turkey was the traditional meat of thanksgiving for the Pilgrims.  It was eaten to celebrate their survival during a winter in Plymouth Colony and it's been associated with the holiday ever since.  The origins of Thanksgiving in the United States can be traced back to the Pilgrims' harvest feast of 1621 that was held in Plymouth Colony, where some of the settlers had gathered to celebrate the good harvests, after three years of hardship.  They celebrated the good luck of the first harvest and the start of a bountiful season with a feast of turkey, beans, corn, and squash.  The feast was held in November in order to commemorate the harvest and celebrate the good fortune of the colony.  In the ensuing years, thanksgiving became an established tradition in the United States.  The holiday was originally celebrated with a turkey dinner, but it soon became an official holiday that celebrated the blessings of the harvest and the reaping of the bounty. |
| **Iterative DPO** | A common misconception about the origins of Thanksgiving is that it was originally a Christian holiday.  However, the roots of Thanksgiving are actually much older than that.  It is likely that the first Thanksgivings were harvest feasts hosted by farmers and local communities.  They celebrated the success of the harvest and the abundance of food available.  Thanksgiving was also likely some sort of harvest celebration, but would have also served as a holiday to celebrate having survived the winter and to give thanks for their farms and their families.  It̆2019s likely that these harvest feasts continued to become more elaborate and elaborate until they became Thanksgiving in 1621. |
| **GPT-4o Judgment** | Response 1 is more helpful because it directly addresses the question by explaining the association of turkey with the Pilgrims 1621 feast and how it became a Thanksgiving tradition, whereas Response 2 is more general and contains inaccuracies and irrelevant details.  More helpful:  1 |

Table 11: GPT-4o chooses iterative PRPO over iterative DPO trained on Pythia6.9b model. Responses to a prompt from the Anthropic-HH test set.

| Prompt | Human:  Should I rent a house or rent an apartment?
Assistant: |
|---|---|
| Iterative PRPO | Both of these options are good, but there are some pros, cons, and other factors to consider when choosing which one to rent. A house can be comfortable and soothing, while an apartment might be more spacious and convenient.  If you like a certain neighborhood or home, it could be more convenient to rent a house and stay with that.  Also, if you're just moving in, or moving between places, you might want to rent an apartment to avoid complications with moving furniture.  Finally, if you2019re planning to stay for a long time, it might be more practical to rent an apartment.  If you're not planning to stay for long, or just want something easy to move into, a house might be more convenient." |
| Iterative DPO | It depends on your goals.  If you want to live, say, in the country or in a city, you may be more likely to rent an apartment.  But if you're just planning to have a place to sleep for a few nights, you might want to rent a house, since you can usually get more privacy with a house than a room in an apartment.  If you're looking to move in the future, you might prefer to rent a house, since it's much easier to rent a house than to move, and you can usually find houses a lot cheaper in the city than in the country. |
| GPT-4o Judgment | Comparison:  Response 1 is more helpful because it clearly outlines practical pros and cons of renting a house versus an apartment, while Response 2 is confusing and contains contradictory or inaccurate claims about location and pricing. More helpful:  1 |

Table 12: GPT-4o chooses iterative PRPO over iterative DPO trained on Pythia6.9b model. Responses to a prompt from the Anthropic-HH test set.

