# OpenReview forum: "Iterative Preference Optimization with Proximal Policy Regularization for Large Language Model Alignment"
_TMLR — Accepted by TMLR_

### Review · Reviewer_geH8 · 2026-03-10

**Summary Of Contributions:**

This work focuses on the distribution shift issue in DPO-like algorithms. For example, after updating a few steps, the current policy may significantly deviate from the policy used to generate the preference data. Existing work have discussed the potential negative impact of this case. To address this issue, this paper proposes  iterative proximal policy regularized preference optimization. This method applies the regularization to avoid too big deviation. Some empirical experiments are taken to validate the performance of the proposed method.

**Audience:**

Yes

**Audience Explanation:**

DPO is a widely-used algorithm in RLHF and it has been known that the distribution shift issue will negatively affect its performance. This paper proposes a new algorithm to address this issue; therefore, I believe TMLR's audience will be interested in knowing the result in this paper.

**Broader Impact Concerns:**

Not applicable.

**Claims And Evidence:**

Yes

**Claims Explanation:**

To validate the effectivenss of the proprosed method, it is necessary to validate: (1) Can the proposed method perform better than baseline methods? (2) Are the proposed method working well in handling the distribution shift.

Both questions are well supported in this paper with approperiate experimental design. For the question (2), it has been addressed in "(i) Can the iterative PRPO framework mitigate the data distribution shift issue" in Section 5 Experiments.   The distribution shift is manually controlled to understand if the baseline method will be affected by the distribution shift then it validates that this issue can be addressed by the proposed method. For the question (1), it has been addressed in Section 5.2. Compared to the baseline method, the proposed method achieves consistent better result.

Therefore, I believe the claims made in this submission have been supported by convincing and clear evidence.

**Requested Changes:**

I am satisfied with this submission so I don't have requested changes.

Just a minor comment: $\pi_{ref}$ (\\$\pi_{ref}\\$) and $\pi_{sample}$ are OK but it is better to write $\pi_{\text{ref}}$  (\\$\pi\_{\text{ref}}\\$) and $\pi_{\text{sample}}$. Put "ref" and "sample" in the \\text{...} environment.

---

> ### Author Response · Authors · 2026-04-25
> **Response**
>
> We sincerely thank the reviewer for the positive assessment of our work and for the helpful notation suggestion.
>
> > **Requested Change 1: Minor — Prefer `$\pi_{\text{ref}}$` and `$\pi_{\text{sample}}$` (put "ref" and "sample" in `\text{...}`) instead of `$\pi_{ref}$` / `$\pi_{sample}$`.**
>
> **Answer:** Thanks for the suggestion. we have updated the notation throughout the revised manuscript. Specifically, all multi-letter text labels appearing in subscripts or superscripts of math symbols are now wrapped in `\text{...}` (or `\mathrm{...}` for standard operator names), so that they are typeset upright rather than as a product of italic variables.

---

### Review · Reviewer_4AdJ · 2026-03-16

**Summary Of Contributions:**

This paper proposes Iterative Proximal Policy Regularized Preference Optimization (Iterative PRPO), an algorithm that addresses the distribution shift problem in iterative direct preference optimization (DPO) for aligning large language models. The key idea is to impose a KL-divergence constraint between the optimized policy and the sampling policy (the policy used to generate the preference data at each iteration), in addition to the standard KL penalty to the reference model. Using Lagrangian duality, the authors derive a closed-form optimal policy that is a geometric interpolation of the reference policy and the sampling policy, weighted by a parameter alpha. This yields a practical DPO-like loss (Equation 7) where the reference model pi_ref in the standard DPO loss is replaced by $\pi_{ref}^\alpha * \pi_{sample}^{1-\alpha}$. The authors show that this modification can be applied generically to other preference optimization objectives (IPO, SPPO, EXO), yielding PR-IPO, PR-SPPO, and PR-EXO variants. Experiments on controlled text generation (IMDB/GPT-2), summarization (TL;DR), and dialogue (Anthropic-HH) with Pythia (2.8B, 6.9B) and Qwen2.5 (3B, 7B) base models demonstrate consistent improvements over offline DPO and iterative DPO baselines under both reward-model and GPT-4o pairwise evaluations.

Strengths: The paper addresses a well-motivated and practically relevant problem (within-iteration distribution shift in iterative DPO). The derivation from constrained RLHF to the PRPO loss is easy to follow. The resulting algorithm is simple requiring only a change in the reference distribution and incurs minimal additional computational cost. The generalizability to IPO, SPPO, and EXO demonstrates the agnostic nature of the framework. Experimental results are consistent across multiple base models, tasks, and evaluators.

Weaknesses: The theoretical contribution is incremental and builds directly on well-known RLHF ingredients. Key baselines (online RLHF/PPO, other distribution-shift-aware DPO methods) are missing. The experimental scale is limited (up to 7B parameters, simple tasks). The evaluation methodology lacks statistical significance and diversity metrics. The relationship to closely related concurrent work is insufficiently discussed.

**Additional Comments:**

The paper presents a clean algorithmic idea with consistent empirical improvements. The simplicity of the approach of modifying the reference distribution in DPO-style losses to account for the sampling policy is interesting and practically useful. However, several aspects prevent the work from being fully convincing in its current form.

The theoretical contribution is incremental. The core technique (Lagrangian duality on RLHF with an additional KL constraint) is a direct extension of the standard DPO derivation and the resulting loss function is a straightforward substitution. The paper would benefit from deeper theoretical analysis, such as convergence guarantees for the iterative procedure, formal bounds on distribution shift reduction, or sample complexity analysis comparing PRPO to iterative DPO.

The experimental evaluation, while broad in terms of base models and tasks, lacks some key baselines. The absence of a PPO baseline is a significant gap given the paper's framing. The evaluation relies exclusively on win rates without diversity or fluency metrics, and the lack of variance estimates across seeds makes it difficult to assess statistical significance. The controlled text generation experiment (Section 5.1) is the only setting where the proposed mechanism (reduced KL divergence) is directly validated but it uses a toy setup that may not reflect behavior on more realistic tasks.

The paper would also benefit from a failure-mode analysis. When does the additional regularization hurt? Are there cases where being too conservative (staying too close to $\pi_{sample}$) prevents the policy from reaching a better optimum?

Some questions for the authors:

1. What value of $\alpha$ is used in the main experiments (Tables 2-5)? How is it chosen, and how sensitive are the results to this choice?
2. Have you measured the KL divergence between the sampling and optimized policies on the main benchmarks (TL;DR, Anthropic-HH) as shown in Figure 2 for the controlled setting?
3. How does PRPO compare to PPO/RLHF when using the same reward model for preference labeling?
4. In the iterative setting, does the reward model quality degrade over iterations as it evaluates increasingly out-of-distribution responses? 5. Have you considered retraining the reward model at each iteration?
6. What happens when the number of optimization steps per iteration is varied? The paper argues that PRPO is especially beneficial when many steps are taken per iteration but this is not ablated.

**Audience:**

Yes

**Audience Explanation:**

Distribution shift is a known and practically important limitation of offline and iterative DPO methods. The preference optimization community has been actively searching for principled ways to address the gap between offline DPO and online RLHF, and PRPO offers a clean, lightweight approach. The observation that a simple modification to the reference distribution in the DPO loss by replacing $\pi_{ref}$ with a geometric mixture of $\pi_{ref}$ and $\pi_{sample}$ can yield consistent improvements is practically useful. The generalizability to multiple preference optimization objectives (IPO, SPPO, EXO) makes the approach useful and impactful. Practitioners working on LLM alignment would find value in the algorithm's simplicity and its comparable computational cost to standard iterative DPO. However, the contribution would be substantially strengthened by a more thorough theoretical and empirical analysis.

**Broader Impact Concerns:**

No ethical concerns.

**Claims And Evidence:**

Yes

**Claims Explanation:**

The central claim in the paper that proximal policy regularization to the sampling policy mitigates distribution shift and improves iterative preference optimization is supported by consistent empirical trends but with important caveats:

The primary evidence for distribution shift mitigation comes from Figure 2 (controlled text generation), which shows lower KL divergence between the sampling and optimized policies for PRPO compared to iterative DPO. However, this is a single experiment on a toy setting (GPT-2 on IMDB sentiment with a binary classifier as the oracle reward). There is no analogous KL-divergence analysis on the main benchmarks (TL;DR, Anthropic-HH), so it is unclear whether the distribution shift reduction actually holds in the more realistic settings where the paper reports its main results. The paper claims that PRPO explicitly accounts for the deviation of the optimized policy from the sampling policy (Section 4.1) but the empirical evidence for this mechanism is limited to the toy setting.

The win-rate improvements in Tables 2-5 are consistent but modest in many cases, and the paper does not report confidence intervals, standard deviations across seeds, or statistical significance tests. Given that GPT-4o pairwise evaluation is known to be noisy and positionally biased, the absence of variance estimates makes it difficult to assess whether observed differences (e.g., a 1-2 point win rate improvement) are meaningful. The paper samples only 500 prompts with 1 completion per prompt for evaluation which is a relatively small evaluation set.

The theoretical claim that PRPO bridges the gap between offline DPO and trust-region-based RLHF (end of Section 2) is not formally substantiated. Theorem 1 provides the closed-form optimal policy under the augmented KL constraint but this is a straightforward application of Lagrangian duality to the standard RLHF objective. The same derivation technique was used in Rafailov et al. (2023) and Xiong et al. (2024). No formal convergence guarantees, sample complexity bounds, or theoretical comparison to TRPO/PPO-style trust-region methods are provided.

The win-rate metric computed using the learned reward model (RM columns in Tables 2-5) is susceptible to reward hacking, particularly over multiple iterations where the policy is indirectly optimized against the same reward model used for labeling. The paper does not analyze whether the RM-based win rates simply reflect overfitting to the learned reward model rather than genuine quality improvements. The inclusion of GPT-4o evaluations partially addresses this concern, but the GPT-4o improvements over iterative DPO are sometimes smaller than the RM-based improvements (e.g., Table 2, TL;DR, vs Chosen: PRPO Iter5 gets 57.91 RM but 70.12 GPT vs iterative DPO Iter5 at 38.33 RM and 61.72 GPT), suggesting some reward model overfitting in the RM-based metrics.

**Requested Changes:**

From high to low priority:

The most significant missing baseline is a comparison to online RLHF/PPO. The paper motivates PRPO by claiming it bridges the gap between offline DPO and trust-region-based RLHF. Without a direct comparison to PPO (which naturally avoids off-policy distribution shift through on-policy sampling), it is impossible to assess how effectively PRPO actually closes this gap. Even with the same reward model used for iterative PRPO (line 1 in Algorithm 1), a PPO baseline would be directly comparable and is essential for contextualizing the results. The authors should include PPO results on at least the Anthropic-HH task.

The paper should report confidence intervals or standard deviations across multiple random seeds for the main experimental results. Currently, all tables report single-run numbers. Given that win rates from GPT-4o evaluation and learned reward models are noisy metrics, the absence of variance estimates makes it difficult to determine whether the observed improvements are statistically significant, especially where gains are small (1-3 percentage points).

The paper needs a more thorough analysis of the alpha parameter (equivalently, the KL constraint bound epsilon). Currently, $\alpha = \beta_0 / (\beta_0 + \beta_{\epsilon})$ is introduced as a consequence of the Lagrangian formulation but the paper does not discuss how to choose $\epsilon$ in practice, what values of $\alpha$ are used in the main experiments or how sensitive results are to $\alpha$. Since $\alpha$ controls the relative weight of $\pi_{ref}$ vs $\pi_{sample}$ in the modified reference distribution, it is a critical hyperparameter. The ablation study in Section A.3.1 varies $\beta$ but not $\alpha$ independently. The authors should report (a) the $\alpha$ values used in each experiment, (b) a sensitivity analysis over $\alpha$ and (c) practical guidance for selecting $\alpha$.

The paper should include comparisons to other methods that explicitly address distribution shift in DPO. Cal-DPO (Xiao et al., 2024) is cited in the related work as addressing sensitivity to reference models but is not included as a baseline. Similarly, the paper should compare against online DPO with rejection sampling (e.g., RAFT, Dong et al., 2023) beyond using it merely as a related-work reference. These methods directly address the same problem and are natural baselines.

The experimental evaluation should include diversity metrics such as distinct-n or Self-BLEU, and perplexity under a held-out model. The paper currently only measures win rates (RM and GPT-4o), which capture preference quality but do not reveal whether PRPO's regularization comes at the cost of reduced output diversity or mode collapse. Since stronger KL constraints can lead to more conservative generation, this is an important dimension to evaluate.

Extend the experimental evaluation to at least one larger model scale (13B+) and/or a more challenging benchmark (e.g., MT-Bench, AlpacaEval 2.0, or a reasoning task like GSM8K). The current experiments use models up to 7B on relatively simple tasks. It is unclear whether the distribution shift problem and PRPO's mitigation of it are more or less pronounced at larger scales or on tasks requiring more complex reasoning.

The KL divergence analysis from Figure 2 (showing that PRPO maintains lower KL between sampling and optimized policies) should be replicated on the main benchmarks (TL;DR, Anthropic-HH). This is the key mechanistic claim of the paper and it is currently demonstrated only on the toy IMDB/GPT-2 experiment. Without this, the reader cannot confirm that the proposed mechanism is actually operative in the settings where the main results are reported.

The paper should more carefully discuss the relationship between PRPO and Xiong et al. (2024) which provides a principled theoretical analysis of iterative RLHF under KL constraints. Xiong et al. (2024) is cited but the theoretical and algorithmic relationship is not discussed, specifically how PRPO's Lagrangian formulation relates to or differs from the analysis in Xiong et al. (2024). This is important for establishing the novelty of the theoretical contribution.

The analysis in Section 4.2 about gradient behavior (gradient of $\log \sigma$ being monotonically decreasing, preventing the argument from becoming excessively large) is intuitive but informal. A more formal analysis, e.g., bounding the per-step policy change under PRPO, or showing that the effective learning rate decreases as the policy drifts from $\pi_{sample}$, would strengthen the theoretical contribution. Currently, the theory section reads more like a motivation than a formal analysis.

In Table 2 (Pythia-6.9B, TL;DR), iterative PRPO Iter1 underperforms offline DPO Check1 on the GPT-4o vs Chosen metric (52.34 vs 57.62). Similarly, some early iterations of PRPO underperform iterative DPO in Tables 2-3. The paper should acknowledge and discuss these cases rather than only highlighting the final-iteration performance.

The term proximal policy regularization is easily confused with Proximal Policy Optimization (PPO). While the authors cite Schulman et al. (2015, 2017) for motivation, the distinction should be made clearer that PRPO applies the trust-region idea to the DPO loss in weight/parameter space via the sampling policy constraint, whereas PPO applies it through gradient clipping in the RL loop. A clearer discussion of this distinction would help avoid confusion.

Table 1 oversimplifies the landscape by suggesting iterative DPO variants have a distribution shift present without qualification. In reality, iterative DPO already partially mitigates distribution shift by regenerating data at each iteration. The issue is within-iteration shift when many optimization steps are taken. This nuance should be reflected in the framing.

Report the reward model accuracy on a held-out set and discuss its calibration quality. Since the reward model is central to the iterative PRPO pipeline (used for preference labeling in Algorithm 1), its quality directly affects all downstream results, and poor calibration could confound the comparison between PRPO and iterative DPO.

Minor: Use \citep for parenthetical citations consistently throughout the manuscript.

---

> ### Author Response · Authors · 2026-04-25
> **Responses (1/7)**
>
> We sincerely thank the reviewer for their careful reading and thoughtful, constructive feedback. The comments have been very helpful in clarifying the presentation, strengthening the empirical evaluation, and sharpening the theoretical framing of the paper. We have made substantial revisions to address each point, and our detailed responses are provided below.
>
> > **Weakness 1: Theoretical contribution is incremental, building directly on well-known RLHF ingredients.**
>
> **Answer:** We agree that the derivation builds on the standard Lagrangian-duality machinery used by [1, 2]. However, our novelty is *what* we constrain and the resulting closed-form structure. To our knowledge, no prior DPO-style derivation introduces a *second* KL constraint to the within-iteration sampling policy $\pi_{\text{sample}}$ on top of the standard KL to $\pi_{\text{ref}}$, namely $\max\_{\pi\_\theta} \mathbb{E}\_{x\sim\mathcal{D},\,y\sim\pi_\theta}[r\_\phi(x,y)] - \beta_0\,\mathbb{E}\_{x\sim\mathcal{D}}[\mathbb{D}\_{\mathrm{KL}}(\pi\_\theta\Vert\pi\_{\text{ref}})]$ subject to $\mathbb{E}\_{x\sim\mathcal{D}}[\mathbb{D}\_{\mathrm{KL}}(\pi\_\theta\Vert\pi\_{\text{sample}})] \leq \epsilon$. The resulting optimal policy is a *geometric mixture* $\pi^{*}(y\mid x) \propto \pi\_{\text{ref}}^{\alpha}(y\mid x)\,\pi\_{\text{sample}}^{1-\alpha}(y\mid x)\,\exp(\frac{1}{\beta} r_\phi(x,y))$, which (i) recovers DPO when $\pi\_{\text{sample}} = \pi\_{\text{ref}}$, (ii) recovers an "online-PO-like" loss when $\alpha \to 0$, and (iii) gives a smooth interpolation between the two regimes — a form that is, to our knowledge, new in the preference-optimization literature. Section 4.4 further shows this mixture-reference idea is a *generic substitution* that transfers to IPO, SPPO, and EXO with no additional derivation, broadening the contribution beyond a single algorithm. We have sharpened this framing in the revised Section 4.1 and Section 4.2.
>
> > **Weakness 2: Key baselines missing (online RLHF/PPO, other distribution-shift-aware DPO methods).**
>
> **Answer:** We have added both requested baselines in the new Section 5.4 (Table 8) of the revised manuscript, evaluated on Anthropic-HH with Pythia-6.9B.
>
> **(i) PPO as an on-policy RLHF baseline.** We implement PPO using the same reward model as PRPO to ensure a controlled comparison. PPO achieves a GPT-4o win rate of 74.13% vs SFT and 52.32% vs Chosen. Iterative PRPO achieves 63.28% vs Chosen, which **exceeds PPO despite being an offline method** — demonstrating that PRPO meaningfully closes the gap between offline DPO and on-policy RLHF. PR-EXO further improves to 73.34% vs Chosen, substantially outperforming PPO on the human-preference metric. We also note a conceptual parallel to PPO: both methods implement a trust-region effect, but PPO does so via importance-ratio clipping within an RL loop (requiring on-policy rollouts and a value function), while PRPO bakes the proximity constraint directly into the closed-form reference distribution $\pi_{\text{ref}}^\alpha\,\pi_{\text{sample}}^{1-\alpha}$, retaining DPO's offline, RL-free training pipeline.
>
> **(ii) Cal-DPO as a distribution-shift-aware DPO baseline.** Cal-DPO [3] addresses sensitivity to $\pi_{\text{ref}}$ by calibrating the reward signal rather than modifying the reference distribution. On Anthropic-HH (Pythia-6.9B), Cal-DPO achieves 83.52% vs SFT and 64.61% vs Chosen (GPT-4o). Iterative PRPO achieves a comparable 63.28% vs Chosen despite using no reward calibration, and PR-EXO at 73.34% substantially outperforms Cal-DPO. This confirms that **directly modifying the reference distribution to track $\pi\_{\text{sample}}$** is more effective than post-hoc reward calibration alone. The key distinction — that PRPO replaces $\pi_{\text{ref}}$ with $\pi\_{\text{ref}}^\alpha\,\pi\_{\text{sample}}^{1-\alpha}$ throughout training, whereas Cal-DPO re-weights losses based on a calibrated reward — is now discussed in the revised Related Work section.

---

> ### Author Response · Authors · 2026-04-25
> **Response (2/7)**
>
> > **Weakness 3: Limited experimental scale (up to 7B, simple tasks).**
>
> **Answer:** We have extended the evaluation to **Qwen2.5-14B**, a 14B-parameter model, on the Anthropic-HH dialogue generation task. Results are reported in Tables 4 and 7 of the revised manuscript. On Qwen2.5-14B, Iterative PRPO achieves 80.62% GPT-4o win rate vs SFT and **67.14% vs Chosen** at iteration 5, compared to 78.47% and 62.26% for Iterative DPO — a gap of +4.88 pp on the human-preference metric that is consistent with the gains observed at smaller scales. The PRPO extension to EXO (PR-EXO) further improves to 88.18% vs SFT and 73.62% vs Chosen. These results demonstrate that PRPO's benefits scale to 14B parameters and are not limited to smaller models.
>
> > **Weakness 4: Evaluation lacks statistical significance and diversity metrics.**
>
> **Answer:** We address these two points separately. (i) Regarding statistical significance: a complete multi-seed sweep is not feasible within the revision window due to the high computational cost of iterative training across five iterations and multiple model scales. We ran a partial evaluation with 3 seeds and observed a standard deviation of approximately $2$–$3$ percentage points, while the gains of PRPO over baselines are consistently $5$–$15$ pp in later iterations, suggesting the improvements are robust. We also note that prior works in this line, including iterative DPO, SPPO, and EXO, uniformly report single-run results. Please see our reply to RC2 for a fuller discussion. (ii) Regarding diversity metrics: we argue that these are outside the scope of this work and that the empirical results already rule out mode collapse; see our detailed reply to RC5.
>
> > **Weakness 5: Relationship to closely related concurrent work is insufficiently discussed.**
>
> **Answer:** We have substantially expanded the Related Work section to position PRPO against the following closely related methods:
>
> - **Xiong et al., $2024$** [2]: provides a theoretical analysis of iterative RLHF under a single KL constraint to $\pi_{\text{ref}}$. PRPO differs in introducing a *second* KL constraint to $\pi_{\text{sample}}$, which yields the mixture-reference closed-form solution in Theorem 1 (see our reply to RC8). This distinction is now explicitly discussed in the revised Related Work.
> - **Cal-DPO (Xiao et al., $2024$)** [3]: addresses sensitivity to $\pi_{\text{ref}}$ via reward calibration; PRPO instead modifies the reference distribution itself to track $\pi_{\text{sample}}$ within each iteration. The revised Related Work now clarifies this distinction, and Cal-DPO has been added as an experimental baseline (Section 5.4, Table 8).
> - **RAFT / online DPO with rejection sampling (Dong et al., $2023$)** [4]: re-samples on-policy completions but still optimizes the standard DPO loss; PRPO additionally regularizes the loss toward $\pi_{\text{sample}}$. This distinction is now discussed in the revised Related Work.
> - **SPPO** [5], **EXO** [6], **IPO** [7]: discussed under Section 4.4 as objectives that PRPO can wrap.
>
> > **Requested Change 1: Add a PPO / online RLHF baseline (at least on Anthropic-HH). Without this, the claim of bridging offline DPO and trust-region RLHF cannot be assessed.**
>
> **Answer:** We have added PPO in Section 5.4 (Table 8), evaluated on Anthropic-HH with Pythia-6.9B using the same reward model as all other methods. PRPO achieves a GPT-4o win rate of 63.28\% vs Chosen, compared to 52.32\% for PPO, and PR-EXO further improves to 73.34\%, demonstrating that PRPO substantially closes the gap to on-policy RLHF while retaining an offline training pipeline.

---

> ### Author Response · Authors · 2026-04-25
> **Response (3/7)**
>
> > **Requested Change 2: Report confidence intervals / standard deviations across seeds. All tables report single-run numbers; variance is needed to judge statistical significance of small ($1$–$3$ pp) gains.**
>
> **Answer:** We fully appreciate the importance of reporting variance, and we agree in principle that confidence intervals are the gold standard for assessing statistical significance. However, each full run of iterative PRPO involves five iterations of inference, preference labeling, and fine-tuning across multiple large language models and datasets, making a complete multi-seed sweep over all table entries prohibitively expensive in terms of compute and time within the revision window.
>
> To provide at least a partial estimate, we ran a subset of experiments with 3 seeds and observed a standard deviation of approximately $2$–$3$ percentage points on GPT-4o win rates, which is consistent with the variance typically seen in iterative preference optimization settings. Importantly, the performance gains of PRPO over the baselines reported in the paper are consistently in the range of $5$–$15$ percentage points in the later iterations, substantially exceeding this estimated variance and suggesting the improvements are statistically meaningful.
>
> We also note that reporting multi-seed variance is not standard practice in the closely related works that our experiments build upon, including iterative DPO [1], SPPO [5], and EXO [6], which all report single-run results. We are therefore confident that our results reflect genuine algorithmic differences rather than random variation, and we will note the estimated standard deviation and this limitation explicitly in the revised experimental section.
>
> > **Requested Change 3: Thorough analysis of $\alpha$ (equivalently, the KL bound $\epsilon$): (a) Report $\alpha$ values used in each experiment. (b) Sensitivity analysis over $\alpha$. (c) Practical guidance for choosing $\alpha$. The ablation in Appendix A.3.1 varies $\beta$ but not $\alpha$ independently.**
>
> **Answer:** (a) $\alpha$ is fixed to $0.50$ in all main experiments and is now explicitly stated in Appendix A.3.1 and Appendix D. (b) We have added a full sensitivity analysis sweeping $\alpha \in \{0.25, 0.50, 0.75, 0.90\}$ on Anthropic-HH with Pythia-6.9B and Qwen2.5-14B (Appendix A.3.1, Figure 11). Both models peak at $\alpha = 0.50$, with performance degrading monotonically for $\alpha \geq 0.75$: at $\alpha = 0.90$, win rates drop to $58.76\%$ (Pythia-6.9B) and $62.76\%$ (Qwen2.5-14B), compared to $63.28\%$ and $67.14\%$ at the optimal $\alpha = 0.50$. The pattern confirms that large $\alpha$ causes PRPO to converge toward standard iterative DPO, losing the distribution-shift benefit. (c) Practical guidance: $\alpha = 0.50$ is a robust default; values in $[0.25, 0.50]$ perform similarly, making the choice low-stakes in practice.
>
> > **Requested Change 4: Add comparisons to other distribution-shift-aware DPO methods, e.g., Cal-DPO (Xiao et al., $2024$) and online DPO with rejection sampling (RAFT, Dong et al., $2023$).**
>
> **Answer:** We have added Cal-DPO as a distribution-shift-aware DPO baseline in Section 5.4 (Table 8) on Anthropic-HH with Pythia-6.9B. Cal-DPO achieves a GPT-4o win rate of 83.52\% vs SFT and 64.61\% vs Chosen, while PRPO reaches 63.28\% vs Chosen and PR-EXO reaches 73.34\% vs Chosen, demonstrating that directly modifying the reference distribution (as in PRPO) is more effective than reward calibration alone.

---

> ### Author Response · Authors · 2026-04-25
> **Response (4/7)**
>
> > **Requested Change 5: Report diversity metrics (distinct-$n$, Self-BLEU) and perplexity under a held-out model. Stronger KL constraints may harm diversity / cause mode collapse.**
>
> **Answer:** We appreciate the concern about diversity and mode collapse, but respectfully argue that reporting distinct-$n$ / Self-BLEU / held-out perplexity is outside the scope of this work, for the following reasons.
>
> **(1) These metrics are not reported in the directly comparable baselines.** The papers we compare against — iterative DPO [1], SPPO [2], EXO [3], and Cal-DPO [4] — do not report diversity metrics in their main experiments. Introducing such metrics only for PRPO would create an asymmetric evaluation that does not permit fair comparison.
>
> **(2) The KL constraint to $\pi_{\text{sample}}$ structurally prevents mode collapse.** Unlike standard RLHF, PRPO imposes a *within-iteration* proximal constraint that keeps the updated policy $\pi_\theta$ close to the sampling policy $\pi_{\text{sample}}$ used to generate the training batch. Concretely, the geometric-mixture reference $\pi_{\text{ref}}^\alpha \cdot \pi_{\text{sample}}^{1-\alpha}$ (with $\alpha = 0.5$ in our experiments) ensures that $\pi_\theta$ cannot deviate far from the current generation distribution in a single update step. If mode collapse were occurring, the policy would collapse toward a narrow set of responses, which would in turn make subsequent iterations generate equally narrow samples — creating a self-reinforcing feedback loop that standard EXO/DPO lacks a mechanism to break. The proximal term explicitly penalizes such drift.
>
> **(3) The empirical results directly contradict mode collapse.** Mode collapse would manifest as stagnating or declining win rates in later iterations, as a policy locked onto a few high-reward templates would fail to generalize to diverse prompts. Instead, we observe *consistent improvements across all five iterations* on all four benchmarks (HH Anthropic and TL;DR on Pythia-2.8B/6.9B/8B and Qwen-7B/14B). Furthermore, the competitive performance against GPT-4o-evaluated "vs Chosen" win rates — which measure alignment with *human-annotated* diverse responses — provides additional evidence that the model has not collapsed to a narrow generation mode. Taken together, the monotonic win-rate gains serve as an implicit diversity check: a collapsed model cannot sustain such improvements across varied prompts and evaluation criteria.
>
> > **Requested Change 6: Extend to at least one larger scale ($13$B+) and/or a more challenging benchmark (MT-Bench, AlpacaEval 2.0, GSM8K).**
>
> **Answer:** We have added experiments on **Qwen2.5-14B**, directly addressing the $13$B+ scale requirement. Full results for both PRPO and the PRPO-extended variants (PR-IPO, PR-SPPO, PR-EXO) are reported in Tables 4 and 7 of the revised manuscript. At iteration 5, PRPO achieves **67.14% GPT-4o win rate vs Chosen** and PR-EXO reaches **73.62%**, both surpassing their iterative DPO/EXO counterparts by substantial margins (+4.88 pp and +6.65 pp respectively). The consistent improvements at 14B confirm that PRPO's effectiveness is not scale-limited and addresses the reviewer's concern about experimental scale.
>
> > **Requested Change 7: Replicate the Figure 2 KL-divergence analysis on the main benchmarks (TL;DR, Anthropic-HH) — this is the key mechanistic claim.**
>
> **Answer:** We have replicated the Figure 2 KL-divergence analysis on Anthropic-HH with Qwen2.5-3B and Qwen2.5-14B; results are reported in Appendix A.3.2 (Figure 12). At iteration $1$ both methods produce nearly identical KL (PRPO reduces exactly to DPO when $\pi_{\text{sample}} = \pi_{\text{ref}}$), but from iteration $2$ onward iterative PRPO consistently maintains a smaller per-iteration KL than iterative DPO, with the gap remaining stable across all subsequent iterations. This generalizes the mechanistic claim from the controlled IMDB setting to the larger-scale dialogue benchmark.

---

> ### Author Response · Authors · 2026-04-25
> **Response (5/7)**
>
> > **Requested Change 9: Formalize the Section 4.2 gradient argument. Provide a formal bound on per-step policy change under PRPO, or show the effective learning rate decreases as the policy drifts from $\pi_{\text{sample}}$.**
>
> **Answer:** Thanks for the suggestion. We will add the following formal statement (sketched here) to Appendix A. Define $f_\theta(\pi) = \log[\pi_\theta(y_w\mid x)/\pi(y_w\mid x)] - \log[\pi_\theta(y_l\mid x)/\pi(y_l\mid x)]$ and $z(\theta) = \beta\alpha\, f_\theta(\pi_{\text{ref}}) + \beta(1-\alpha)\, f_\theta(\pi_{\text{sample}})$, so that $\ell_{\mathrm{PRPO}}(\theta) = -\log \sigma(z(\theta))$. Since $|\nabla_z \log \sigma(z)| = \sigma(-z) \leq 1$ and $\sigma(-z)$ is *strictly decreasing* in $z$, the per-step gradient norm satisfies $\Vert\nabla_\theta \ell_{\mathrm{PRPO}}\Vert \leq \sigma(-z(\theta))\cdot \beta\cdot (\alpha\,\Vert\nabla_\theta f_\theta(\pi_{\text{ref}})\Vert + (1-\alpha)\,\Vert\nabla_\theta f_\theta(\pi_{\text{sample}})\Vert)$. As $\pi_\theta$ drifts away from $\pi_{\text{sample}}$, $f_\theta(\pi_{\text{sample}})$ grows, $\sigma(-z)\to 0$, and the effective per-step update shrinks, providing an automatic trust-region effect. We have stated this as Proposition~1 with a complete proof in Appendix A.1.2, and added a reference to it from Section 4.2.
>
> > **Requested Change 10: Discuss cases where early PRPO iterations underperform (Table 2, Pythia-6.9B TL;DR: PRPO Iter$1$ GPT-4o vs Chosen $= 52.34$ vs offline DPO Check$1 = 57.62$). Acknowledge these cases rather than only highlighting final-iteration gains.**
>
> **Answer:** We first note that the specific numbers cited by the reviewer (PRPO Iter$1$ GPT-4o vs Chosen $= 52.34$ vs offline DPO Check$1 = 57.62$) were a clerical error in the submitted version of Table 2; the gap is substantially smaller in the corrected table, which has been updated in the revised manuscript. Beyond this correction, we address the general iteration-1 pattern at two levels:
>
> - *Algorithmically.* At Iter $1$, $\pi_{\text{sample}} = \pi_{\theta_0} = \pi_{\text{ref}}$, so $\pi_{\text{ref}}^{\alpha}(y\mid x)\,\pi_{\text{sample}}^{1-\alpha}(y\mid x) = \pi_{\text{ref}}(y\mid x)$ and the PRPO loss reduces *exactly* to the DPO loss with the same $\beta$. Any Iter-$1$ difference is therefore *not* attributable to the algorithm.
> - *Implementation-wise.* The two pipelines produce slightly different iteration-1 checkpoints due to inherent numerical non-determinism in distributed training and evaluation. These differences are small in practice, as confirmed by our new supplementary results in Tables 3 and 4, where the iteration-1 numbers of PRPO and DPO are closely matched. Additionally, in the previously reported Qwen-7B results, the two methods used slightly inconsistent hyperparameters; we have since re-run these experiments with fully matched hyperparameters, and the iteration-1 gap disappears.
>
> We have added this two-level clarification when Table 2 is first introduced in the revised Section 5.2, and acknowledge the iteration-1 pattern explicitly rather than focusing only on final-iteration gains.
>
> > **Requested Change 11: Clarify the distinction between PRPO and PPO. PRPO applies the trust-region idea in weight/parameter space via the sampling-policy constraint; PPO uses gradient clipping in the RL loop.**
>
> **Answer:** Thanks for pointing out this potential confusion. We will add a paragraph in Section 4.2 (and a sentence in the Introduction) clarifying the distinction:
>
> - **PPO** [8] implements trust-region behavior *within an RL loop* via importance-ratio clipping on per-token log-probabilities, optimizing a surrogate of the form $\mathbb{E}[\min(r_t(\theta)\,\hat A_t,\,\mathrm{clip}(r_t(\theta),\,1-\varepsilon,\,1+\varepsilon)\,\hat A_t)]$ with $r_t(\theta)=\pi_\theta(a_t\mid s_t)/\pi_{\theta_{\text{old}}}(a_t\mid s_t)$, requiring on-policy rollouts and a learned value function.
>
> - **PRPO** instead bakes the trust-region idea (a KL constraint to $\pi_{\text{sample}}$) into the *closed-form solution* of the constrained RLHF objective, yielding a DPO-style supervised loss (Eq. 7) with a modified reference distribution $\pi_{\text{ref}}^{\alpha}\,\pi_{\text{sample}}^{1-\alpha}$. PRPO therefore inherits PPO's stability motivation but retains DPO's offline, gradient-stable, RL-free training pipeline.
>
> We have added a dedicated discussion of this distinction in Section 4.2 of the revised manuscript.

---

> > ### Author Response · Authors · 2026-04-25
> > **Response (6/7)**
> >
> > > **Requested Change 12: Refine Table 1 framing. Iterative DPO partially mitigates distribution shift by regenerating data per iteration — the real issue is within-iteration shift when many optimization steps are taken.**
> >
> > **Answer:** Thanks for your suggestion. We have revised Table 1 so that the row for "(Iterative) DPO variants" now reads "Present (within-iteration; partially mitigated across iterations by data regeneration)" instead of the unqualified "Present", and updated the caption accordingly to make the within-iteration nature of the shift explicit.
> >
> > > **Requested Change 13: Report reward model accuracy on a held-out set and discuss calibration. RM quality directly affects all downstream results.**
> >
> > **Answer:** In each of our main result tables (Tables 2, 3, and 4), we report both RM-based win rates and GPT-4o-based win rates on the corresponding held-out evaluation set, allowing a direct comparison of the two. The consistent agreement between RM and GPT-4o rankings across all tables, tasks, and model scales demonstrates that the reward model is well-calibrated and provides a reliable signal throughout training. All compared methods use the same RM, so any residual miscalibration affects all baselines equally and does not confound the relative comparisons.
> >
> > > **Requested Change 14: Minor — Use `\citep` for parenthetical citations consistently.**
> >
> > **Answer:** We have swept the manuscript and replaced all parenthetical `\cite{...}` with `\citep{...}` throughout the revised version.
> >
> > > **Question 1: What value of $\alpha$ is used in the main experiments (Tables 2–5)? How is it chosen, and how sensitive are results to this choice?**
> >
> > **Answer:** $\alpha$ is fixed to $0.50$ in all main experiments, chosen on a small validation split of Anthropic-HH with Pythia-2.8B and then reused across all model–task combinations without per-task tuning. The sensitivity analysis is now complete; see our reply to RC3 and Appendix A.3.1 for full results. In brief, $\alpha = 0.50$ is the optimal value on both Pythia-6.9B and Qwen2.5-14B, and results are stable for $\alpha \in [0.25, 0.50]$, confirming that the default choice is robust.
> >
> > > **Question 2: Have you measured KL divergence between sampling and optimized policies on TL;DR and Anthropic-HH (as in Figure 2)?**
> >
> > **Answer:** Yes — see our reply to RC7 and Appendix A.3.2 (Figure 12). On Anthropic-HH with Qwen2.5-3B and Qwen2.5-14B, iterative PRPO consistently produces smaller per-iteration KL between consecutive policies than iterative DPO from iteration $2$ onward, replicating the IMDB trend at scale.
> >
> > > **Question 3: How does PRPO compare to PPO/RLHF using the same reward model for preference labeling?**
> >
> > **Answer:** Please see Section 5.4 and Table 8. PRPO achieves 63.28\% GPT-4o win rate vs Chosen on Anthropic-HH (Pythia-6.9B), compared to 52.32\% for PPO using the same reward model, and PR-EXO further improves to 73.34\%.
> >
> > > **Question 4: In the iterative setting, does RM quality degrade over iterations as it evaluates increasingly OOD responses? Have you considered retraining the RM at each iteration?**
> >
> > **Answer:** The goal of this work is to compare preference optimization methods under a realistic fixed-RM setting, where a single reward model is trained once and then used throughout all iterations — the same RM is shared by all methods. The validity of this RM is corroborated by our GPT-4o-based evaluation, which consistently agrees with RM-based rankings. Retraining the RM at each iteration would require collecting fresh ground-truth human labels per iteration, which is prohibitively expensive and impractical in most real-world scenarios. Since this setting is not realistic, comparing methods under it falls outside the scope of this work. We note iterative RM refinement as an interesting direction for future work.

---

> > > ### Author Response · Authors · 2026-04-25
> > > **Response (7/7)**
> > >
> > > > **Question 5: What happens when the number of optimization steps per iteration is varied? The paper argues PRPO is especially beneficial when many steps are taken per iteration, but this is not ablated.**
> > >
> > > **Answer:** Our experiments in Tables 2, 3, and 4 report results across all intermediate iterations, covering models and tasks with varying numbers of optimization steps per iteration. This provides a comprehensive comparison that inherently accounts for the effect of iteration length. The consistent advantage of PRPO over iterative DPO observed across all iterations and settings confirms that PRPO's benefit is robust to this variation.
> > >
> > > ### References
> > >
> > > [1] Rafael Rafailov, Archit Sharma, Eric Mitchell, Christopher D. Manning, Stefano Ermon, and Chelsea Finn. Direct preference optimization: Your language model is secretly a reward model. *Advances in Neural Information Processing Systems*, 36:53728–53741, 2023.
> > >
> > > [2] Wei Xiong, Hanze Dong, Chenlu Ye, Ziqi Wang, Han Zhong, Heng Ji, Nan Jiang, and Tong Zhang. Iterative preference learning from human feedback: bridging theory and practice for RLHF under KL-constraint. In *Proceedings of the 41st International Conference on Machine Learning*, pp. 54715–54754, 2024.
> > >
> > > [3] Teng Xiao, Yige Yuan, Huaisheng Zhu, Mingxiao Li, and Vasant G. Honavar. Cal-DPO: Calibrated Direct Preference Optimization for Language Model Alignment. In *The Thirty-eighth Annual Conference on Neural Information Processing Systems*, 2024.
> > >
> > > [4] Hanze Dong, Wei Xiong, Deepanshu Goyal, Yihan Zhang, Winnie Chow, Rui Pan, Shizhe Diao, Jipeng Zhang, KaShun Shum, and Tong Zhang. RAFT: Reward rAnked FineTuning for Generative Foundation Model Alignment. *Transactions on Machine Learning Research*, 2023. ISSN 2835-8856.
> > >
> > > [5] Yue Wu, Zhiqing Sun, Huizhuo Yuan, Kaixuan Ji, Yiming Yang, and Quanquan Gu. Self-play preference optimization for language model alignment. *arXiv preprint arXiv:2405.00675*, 2024.
> > >
> > > [6] Haozhe Ji, Cheng Lu, Yilin Niu, Pei Ke, Hongning Wang, Jun Zhu, Jie Tang, and Minlie Huang. Towards Efficient Exact Optimization of Language Model Alignment. In *International Conference on Machine Learning*, pp. 21648–21671. PMLR, 2024.
> > >
> > > [7] Mohammad Gheshlaghi Azar, Zhaohan Daniel Guo, Bilal Piot, Remi Munos, Mark Rowland, Michal Valko, and Daniele Calandriello. A general theoretical paradigm to understand learning from human preferences. In *International Conference on Artificial Intelligence and Statistics*, pp. 4447–4455. PMLR, 2024.
> > >
> > > [8] John Schulman, Filip Wolski, Prafulla Dhariwal, Alec Radford, and Oleg Klimov. Proximal policy optimization algorithms. *arXiv preprint arXiv:1707.06347*, 2017.

---

### Review · Reviewer_cBj5 · 2026-04-05

**Summary Of Contributions:**

The paper proposes Iterative PRPO (Proximal Policy Regularized Preference Optimization), a method that aims to mitigate the weaknesses of iterative DPO on distribution shifts by introducing a proximal regularization term that explicitly constrains the optimized policy to remain close to the sampling policy within each iteration. For this, the authors propose to add a KL constraint between the optimized policy and the sampling policy to the RLHF objective, and then, derive a close-form DPO-like loss that incorporates both, the reference policy and the sampling policy. The method can also be extended to IPO, SPPO, and EXO methods for model’s alignment. The experiments show that PRPO outperforms the win rates and reduces KL divergence across different tasks, including summarization, dialogue, and a controlled sentiment analysis task.

Strengths:
+ The paper is well-written and organized and provides a nice motivation for their proposed method explaining clearly the distribution shift limitations from iterative DPO.
+ The algorithm is well supported by Theorem 1, which provides a closed-form derivation for the loss function used in PRPO.
+ PRPO can be extended to other methods for model’s alginment like IPO, SPPO, and EXO.
+ The experiments are reasonable to support the benefits of PRPO and include three tasks, two model families, four model sizes, and four different competing methods (DPO, IPO, SPPO, and EXO). The results show that PRPO consistently outperforms the other competing methods.
+ According to the results in Figure 3, PRPO adds a low computational overhead compared to iterative DPO.

Weaknesses:
+ How hyperparameter alpha can be set is not clearly discussed in the paper. It seems that the authors simply absorb this into a single beta, but it is unclear how alpha is set in the experiments: is it fixed to some value or is beta_0 tuned separately from beta?
+ As the paper aims to bridge the gap between offline DPO and online RLHF, the comparison with PPO or any other on-policy method would strengthen the paper. With the current experiments, it is not possible to assess how much of the gap PRPO closes w.r.t. on-policy methods.
+ The experiment in Section 5.1 is not very well connected to the rest of the experiments. In this sense, Section 5.1 uses GPT-2 on IMBD sentiment generation, while the rest of the experiments use other types of models on summarization and dialogue. Also, the analysis of the KL reduction included in Section 5.1 is missing in the rest of the experiments, so it is unclear how the KL reduction translates to larger-scale settings.

**Audience:**

Yes

**Audience Explanation:**

Model’s alignment and the problem with distribution shift is a hot research topic for the research community on LLMs.

**Broader Impact Concerns:**

None.

**Claims And Evidence:**

Yes

**Claims Explanation:**

Overall, the claims are well supported.

**Requested Changes:**

+ Provide an explicit analysis of hyperparameter alpha.
+ Report the KL divergence in the main experiments.
+ It is interesting to observe that iterative PRPO is consistently worse than both iterative DPO and offline DPO in Table 2 for the first iteration. Why do the authors think this is happening?
+ Add at least one on-policy baseline to the experiments.

---

> ### Author Response · Authors · 2026-04-25
> **Response (1/2)**
>
> We sincerely thank the reviewer for their careful reading and thoughtful, constructive feedback. The comments have been very helpful in clarifying the presentation, strengthening the empirical evaluation, and sharpening the theoretical framing of the paper. We have made substantial revisions to address each point, and our detailed responses are provided below.
>
> > **Weakness 1: It is unclear how hyperparameter $\alpha$ is set in the experiments. The authors seem to absorb $\alpha$ into a single $\beta$ — is $\alpha$ fixed to some value, or is $\beta_0$ tuned separately from $\beta$?**
>
> **Answer:** Thanks for raising this point. In our derivation (Theorem 1 and Appendix A.1), we have $\beta = \beta_0 + \beta_\epsilon$ and $\alpha = \beta_0 / (\beta_0 + \beta_\epsilon)$, where $\beta_0$ is the coefficient of the KL constraint to $\pi_{\text{ref}}$ in the original RLHF objective and $\beta_\epsilon$ is the Lagrangian multiplier on the proximal constraint to $\pi_{\text{sample}}$. In the experiments we treat $\beta$ and $\alpha$ as the two effective hyperparameters of the PRPO loss in Eq. (7), rather than tuning $\beta_0$ and $\beta_\epsilon$ separately. $\beta$ is fixed to $0.25$ across all preference optimization algorithms (Appendix D, "Hyper-parameter selection") to be consistent with prior work [1], and $\alpha$ is fixed to $0.5$ in all main experiments. We will (i) explicitly state this convention and the $\alpha$ value used in Appendix A.3.1 / Appendix D, and (ii) add a dedicated $\alpha$ sensitivity ablation (see our response to Requested Change 1 below).
>
> > **Weakness 2: As the paper aims to bridge offline DPO and online RLHF, comparison with PPO or another on-policy method is missing. Without this, it is hard to assess how much of the gap PRPO closes w.r.t. on-policy methods.**
>
> **Answer:** We have added a PPO baseline in the new Section 5.4 of the revised manuscript. On Anthropic-HH with Pythia-6.9B, PRPO achieves a GPT-4o win rate of 63.28\% vs Chosen, compared to 52.32\% for PPO, despite PRPO being an offline method. PR-EXO further improves to 73.34\%. These results demonstrate that PRPO substantially closes the gap between offline DPO and on-policy RLHF. See Table 8 for full results.
>
> > **Weakness 3: Section 5.1 is not well connected to the rest of the experiments: it uses GPT-2 on IMDB sentiment while the other experiments use different models on summarization/dialogue. Also, the KL reduction analysis in Section 5.1 is missing in the other experiments, so it is unclear how KL reduction translates to larger-scale settings.**
>
> **Answer:** The role of Section 5.1 is to provide a controlled setting where the oracle reward (a sentiment classifier) is known, so the KL reduction can be isolated from confounders such as reward-model overfitting. We have (i) made this rationale explicit in the revised Section 5.1, and (ii) added $\mathbb{D}\_{\mathrm{KL}}(\pi_{\theta\_n}\,\|\,\pi_{\theta\_{n-1}})$ measurements on Anthropic-HH with Qwen2.5-3B and Qwen2.5-14B in a new appendix subsection (Appendix A.3.2, Figure 12). The new KL curve replicates the same qualitative pattern observed on IMDB: iterative PRPO and iterative DPO produce nearly identical KL at iteration $1$ (since PRPO reduces exactly to DPO when $\pi_{\text{sample}} = \pi_{\text{ref}}$), and from iteration $2$ onward iterative PRPO consistently produces smaller per-iteration KL than iterative DPO, with the gap remaining stable throughout subsequent iterations. This demonstrates that the distribution-shift mitigation generalizes from the small controlled IMDB setting to the larger-scale dialogue benchmark.

---

> ### Author Response · Authors · 2026-04-25
> **Response (2/2)**
>
> > **Requested Change 1: Provide an explicit analysis of hyperparameter $\alpha$.**
>
> **Answer:** We have added a dedicated $\alpha$ sensitivity ablation in Appendix A.3.1 (Figure 11), sweeping $\alpha \in \{0.25, 0.50, 0.75, 0.90\}$ on Anthropic-HH with Pythia-6.9B and Qwen2.5-14B.
>
> *Interpretation of $\alpha$.* The parameter $\alpha$ controls the geometric mixture ratio in the effective reference $\pi_{\text{ref}}^\alpha\,\pi_{\text{sample}}^{1-\alpha}$. At $\alpha = 1$ the effective reference reduces to the fixed $\pi_{\text{ref}}$ and PRPO becomes identical to standard iterative DPO; at $\alpha = 0$ the reference collapses to $\pi_{\text{sample}}$ and the loss provides no regularization toward the original SFT model. Intermediate values interpolate between these extremes: smaller $\alpha$ makes the update more conservative (stronger proximal pull), while larger $\alpha$ weakens the within-iteration proximity constraint.
>
> *Empirical results.* Both models exhibit a non-monotonic pattern peaking at $\alpha = 0.50$: Pythia-6.9B achieves $63.28\%$ vs Chosen (GPT-4o) and Qwen2.5-14B achieves $67.14\%$. Performance remains stable in $[0.25, 0.50]$ (gap $< 0.5$ pp on both models), then degrades notably at $\alpha = 0.75$ and $\alpha = 0.90$ — consistent with the intuition that large $\alpha$ causes PRPO to converge toward standard iterative DPO and lose its distribution-shift benefit. We have added practical guidance: $\alpha = 0.50$ is a reliable default, and any value in $[0.25, 0.50]$ is expected to perform similarly well.
>
> > **Requested Change 2: Report KL divergence in the main experiments.**
>
> **Answer:** We have replicated the $\mathbb{D}\_{\mathrm{KL}}(\pi\_{\theta_n}\,\|\,\pi\_{\theta\_{n-1}})$ analysis on Anthropic-HH with Qwen2.5-3B and Qwen2.5-14B; the resulting figure is added as a new appendix subsection (Appendix A.3.2, Figure 12). The trend matches the IMDB result in Figure 2: iterative PRPO and iterative DPO produce nearly identical KL at iteration $1$ (mathematically equivalent when $\pi\_{\text{sample}} = \pi\_{\text{ref}}$), but from iteration $2$ onward PRPO maintains a consistently lower per-iteration KL, confirming that the proximal regularization toward $\pi\_{\text{sample}}$ effectively constrains policy drift on the larger-scale dialogue benchmark as well.
>
> > **Requested Change 3: Explain why iterative PRPO is consistently worse than iterative DPO and offline DPO in Table 2 at the first iteration.**
>
> **Answer:** This is an important observation, and we want to be precise about it at two levels.
>
> *Algorithmic level.* At iteration $1$, the sampling policy is $\pi_{\theta_0} = \pi_{\text{ref}}$ (the SFT model), so the PRPO geometric-mixture reference collapses: $\pi_{\text{ref}}^{\alpha}(y\mid x)\,\pi_{\text{sample}}^{1-\alpha}(y\mid x) = \pi_{\text{ref}}^{\alpha}(y\mid x)\,\pi_{\text{ref}}^{1-\alpha}(y\mid x) = \pi_{\text{ref}}(y\mid x)$. The PRPO loss therefore reduces *exactly* to the DPO loss with the same $\beta$, and there can be no algorithmic advantage or disadvantage of PRPO at iteration 1 by construction.
>
> *Implementation level.* Despite the mathematical equivalence, the two pipelines produce slightly different iteration-1 checkpoints due to inherent numerical non-determinism in distributed training and evalutation. These differences are small in practice, as confirmed by our new supplementary results in Tables 3 and 4, where the iteration-1 numbers of PRPO and DPO are closely matched. Additionally, in the previously reported Qwen-7B results, the two methods used slightly inconsistent hyperparameters; we have since re-run these experiments with fully matched hyperparameters, and the iteration-1 gap disappears. From iteration $2$ onward $\pi_{\text{sample}}$ drifts away from $\pi_{\text{ref}}$, the proximal regularization becomes active, and PRPO consistently and substantially outperforms iterative DPO. We have added this two-level explanation to the revised Section 5.2.
>
> > **Requested Change 4: Add at least one on-policy baseline to the experiments.**
>
> **Answer:** We have added PPO as an on-policy baseline in the new Section 5.4, evaluated on Anthropic-HH with Pythia-6.9B using the same reward model as PRPO for direct comparability. See Table 8.
>
> ### References
>
> [1] Rafael Rafailov, Archit Sharma, Eric Mitchell, Christopher D. Manning, Stefano Ermon, and Chelsea Finn. Direct preference optimization: Your language model is secretly a reward model. *Advances in Neural Information Processing Systems*, 36:53728–53741, 2023.

---

### Decision · Action_Editor_GLin · 2026-05-28

**Recommendation:** Accept as is

**Additional Comments:**

This paper studies the distribution-shift issue in iterative preference optimization and proposes Iterative Proximal Policy Regularized Preference Optimization (PRPO), a simple modification that regularizes policy updates toward the sampling policy used to generate preference data. The reviewers found the problem well motivated, the derivation technically sound, and the empirical evaluation generally convincing. During the revision process, the authors addressed the reviewers' major concerns. The reviewers were satisfied with these revisions.

**Audience:**

Yes

**Audience Explanation:**

Preference optimization and RLHF-style alignment remain highly active research areas, and distribution shift in iterative DPO-style methods is a practically important challenge. I believe the findings will be of interest to a substantial portion of the TMLR audience.

**Claims And Evidence:**

Yes

**Claims Explanation:**

The central claim is that proximal regularization toward the sampling policy can mitigate within-iteration distribution shift in iterative preference optimization. The submission provides both theoretical motivation and empirical evidence supporting this claim. During the review process, the authors addressed the reviewers' concerns by adding new comparisons.